# The biogeographic differentiation of algal microbiomes in the upper ocean from pole to pole

Kara Martin[1,2,11], Katrin Schmidt[3,11], Andrew Toseland [3], Chris A. Boulton [4], Kerrie Barry[5], Bánk Beszteri [6], Corina P. D. Brussaard[7], Alicia Clum[5], Chris G. Daum [5], Emiley Eloe-Fadrosh [5], Allison Fong[8], Brian Foster[5], Bryce Foster[5], Michael Ginzburg [8], Marcel Huntemann [5], Natalia N. Ivanova [5], Nikos C. Kyrpides [5], Erika Lindquist[5], Supratim Mukherjee [5], Krishnaveni Palaniappan[5], T. B. K. Reddy[5], Mariam R. Rizkallah[8], Simon Roux[5], Klaas Timmermans[7], Susannah G. Tringe [5], Willem H. van de Poll [9], Neha Varghese[5], Klaus U. Valentin[8], Timothy M. Lenton [4], Igor V. Grigoriev [5,10], Richard M. Leggett [2], Vincent Moulton[1] & Thomas Mock [3✉]

Eukaryotic phytoplankton are responsible for at least 20% of annual global carbon fixation. Their diversity and activity are shaped by interactions with prokaryotes as part of complex microbiomes. Although differences in their local species diversity have been estimated, we still have a limited understanding of environmental conditions responsible for compositional differences between local species communities on a large scale from pole to pole. Here, we show, based on pole-to-pole phytoplankton metatranscriptomes and microbial rDNA sequencing, that environmental differences between polar and non-polar upper oceans most strongly impact the large-scale spatial pattern of biodiversity and gene activity in algal microbiomes. The geographic differentiation of co-occurring microbes in algal microbiomes can be well explained by the latitudinal temperature gradient and associated break points in their beta diversity, with an average breakpoint at 14 °C ± 4.3, separating cold and warm upper oceans. As global warming impacts upper ocean temperatures, we project that break points of beta diversity move markedly pole-wards. Hence, abrupt regime shifts in algal microbiomes could be caused by anthropogenic climate change.

---

[1] School of Computing Sciences, University of East Anglia, Norwich Research Park, Norwich, UK. [2] Earlham Institute, Norwich Research Park, Norwich, UK. [3] School of Environmental Sciences, University of East Anglia, Norwich Research Park, Norwich, UK. [4] Global Systems Institute, University of Exeter, Exeter, UK. [5] U.S. Department of Energy Joint Genome Institute, Lawrence Berkeley National Laboratory, Berkeley, CA, USA. [6] Department of Biology, University of Duisburg-Essen, Essen, Essen, Germany. [7] Royal Netherlands Institute for Sea Research, Texel, The Netherlands. [8] Alfred Wegener Institute for Polar and Marine Research, Bremerhaven, Germany. [9] Centre for Isotope Research - Oceans, Energy and Sustainability Research Institute Groningen, Faculty of Science and Engineering, University of Groningen, AG Groningen, The Netherlands. [10] Plant and Microbial Biology Department, University of California, Berkeley, CA, USA. [11] These authors contributed equally: Kara Martin, Katrin Schmidt. ✉email: t.mock@uea.ac.uk

Phytoplankton are a diverse group of largely photo-autotrophic microorganisms encompassing algae and cyanobacteria[1,2], contributing approximately half of the annual global carbon fixation[3]. Although the interconnected oceans generally do not limit their global dispersal[4–6] many studies have shown that their local diversity is correlated with geographical partitioning based on either oceanographic fronts that separate populations or larger-scale ecosystem gradients such as the latitude gradient in local species diversity[7–10]. However, there is also evidence that environmental and ecological selection in geographically well-defined and seemingly unstructured marine ecosystems likely plays a role in generating and maintaining microbial diversity[11]. Regardless as to whether inter or intra-specific variations are being considered to explain microbial diversity patterns in the global ocean, two variables usually explain most of the relatedness between species and populations, respectively: temperature and whole-community chlorophyll *a*[9,11]. Temperature is known to be a strong selecting agent evidenced by thermal tolerance limits according to the geographic origin of species[9,12,13]. Furthermore, temperature together with salinity and the flow of currents creates ecological boundaries in the upper ocean such as oceanographic fronts, which might impact the structure and evolution of inter and intra-specific diversity across spatio-temporal scales[10,14]. Chlorophyll *a* on the other hand, which is a proxy for the biomass of phytoplankton, suggests that ecological selection is at play via interactions with organisms that benefit from phytoplankton and vice versa[11]. Besides herbivores such as copepods and krill, heterotrophic microbes such as bacteria and archaea are among those groups with significant interactions with phytoplankton[15]. Some of them even form intimate relationships including mutualism and symbiosis[16,17]. The space where most of the interactions between phytoplankton and heterotrophic prokaryotes take place is the phycosphere, a microscale mucus region that is rich in organic matter surrounding a phytoplankton cell analogous to the rhizosphere in plants[18,19]. Thus, organic matter released by phytoplankton are used as substrates for prokaryotes, which sometimes provide essential bioactive compounds in return, such as vitamin B12. About 60% of examined heterokont microalgae (e.g. diatoms) require vitamin B12 that is synthesized by bacteria and archaea[20]. Thus, those bacteria have formed a mutualistic relationship with phytoplankton that potentially help to sustain primary productivity in many parts of the global ocean[16]. There is also evidence for species-specific diversity of algal microbiomes. Often, it is the phytoplankton partner that recruits heterotrophic microbes via the secretion of infochemicals, which elicits a response from the other microbes[19]. As these signalling processes can be species-specific and likely have co-evolved in association with responding partners, algal microbiomes are complex and dynamic and their diversity might be either driven by ecological or environmental selection, generating and maintaining these intimate relationships over space and evolutionary time.

As algal microbiomes underpin some of the largest food webs on Earth and drive global biogeochemical cycles, significant international efforts, especially over the last decade have provided insights into what drives their diversity and global biogeography in the global ocean. For instance, large-scale ocean omics studies in the epipelagic realm as part of the Tara Oceans project[21,22] showed that associations among microbes were non-randomly distributed in co-occurrence networks and that their structure was driven by both local and global patterns[15]. Microbial networks that included a significant amount of prokaryotic phytoplankton (cyanobacteria) even appear to be responsible for the majority of carbon exported in the oligotrophic ocean[23]. Interestingly, some of the co-occurrence networks that contained eukaryotic phytoplankton groups were not taxon-specific and dominated by mutual exclusions, which suggests that their biogeography may be influenced by predator-prey dynamics[24].

These studies have provided a step change in our understanding of how ecological interactions in the context of changing environmental conditions likely influence the diversity of the photoautotrophic microbial interactome in the global ocean. However, to assess how environmental conditions such as temperature and variable nutrient concentrations impact the diversity of algal microbiomes, it is instrumental to include polar oceans. With their inclusion, the complete spectrum of environmental parameters that co-vary can be used to assess how these parameters on a truly global scale from pole-to-pole impact differences in the variation of species identities and abundances between local assemblages across larger regions (beta diversity)[25,26] of interacting algal microbiomes, which, to the best of our knowledge, has not been addressed in previous studies. The application of beta diversity enables us to understand the degree of differentiation among biological communities, which across the complete latitudinal scale from pole to pole will provide insights into how marine microbes are latitudinally distributed. As the Arctic and Southern Oceans and specifically their eukaryotic phytoplankton and associated prokaryotes are often not included in global biodiversity surveys, our understanding of how environmental variables including habitat characteristics of polar oceans influence differences in their diversity and activity is incomplete. However, with the inclusion of polar communities, biogeographic differentiation will not likely reveal drivers responsible for small-scale and local differences in the relatedness of communities because the extreme ends of the environmental spectrum are being considered. Rather, this approach will provide insights into environmental variables that are likely responsible for the most latitudinal differentiation of microbial diversity, potentially overshadowing variables responsible for local differences in microbial diversity patterns. Our study, therefore, addresses how large-scale environmental differences on a nearly complete latitudinal scale from pole-to-pole correlate with the biogeographical differentiation of algal microbiomes including the gene activity of eukaryotic phytoplankton. Furthermore, as the upper ocean is experiencing significant warming due to the production of anthropogenic carbon dioxide, we estimate how their biogeographic differentiation might alter based on a model from the IPCC 5th Assessment Report. The main outcome of our work shows that physico-chemical differences between polar and non-polar upper oceans have a strong influence on the dissimilarity of algal microbiomes with respect to changes in the diversity of their co-occurring microbes but also the gene expression activity of their primary producers. These results suggest that there is an ecological boundary in sub-polar oceans of both hemispheres, which not only alters the spatial scaling of algal microbiomes but also shifts pole-wards due to global warming.

## Results

### A meta-omics resource for algal microbiomes in the upper ocean from pole to pole

Three different omics datasets were collected for this study from chlorophyll *a* maximum layers of the Arctic, Atlantic and Southern Oceans (Fig. 1A): (1) 79× eukaryotic metatranscriptomes, 2) 57 × 16S and (3) 54 × 18S rDNA amplicon (V4 region) datasets as subsets of the 82 total samples (Fig. 1A). Sequencing was done at the U.S. Department of Energy Joint Genome Institute (JGI) as part of the JGI Community Science Project 532/300780 (Sea of Change: Eukaryotic Phytoplankton Communities in the Arctic Ocean).

This dataset consists of sequence data from 4 separate cruises: ARK-XXVII/1 (PS80)—17th June to 9th July 2012; Stratiphyt-II

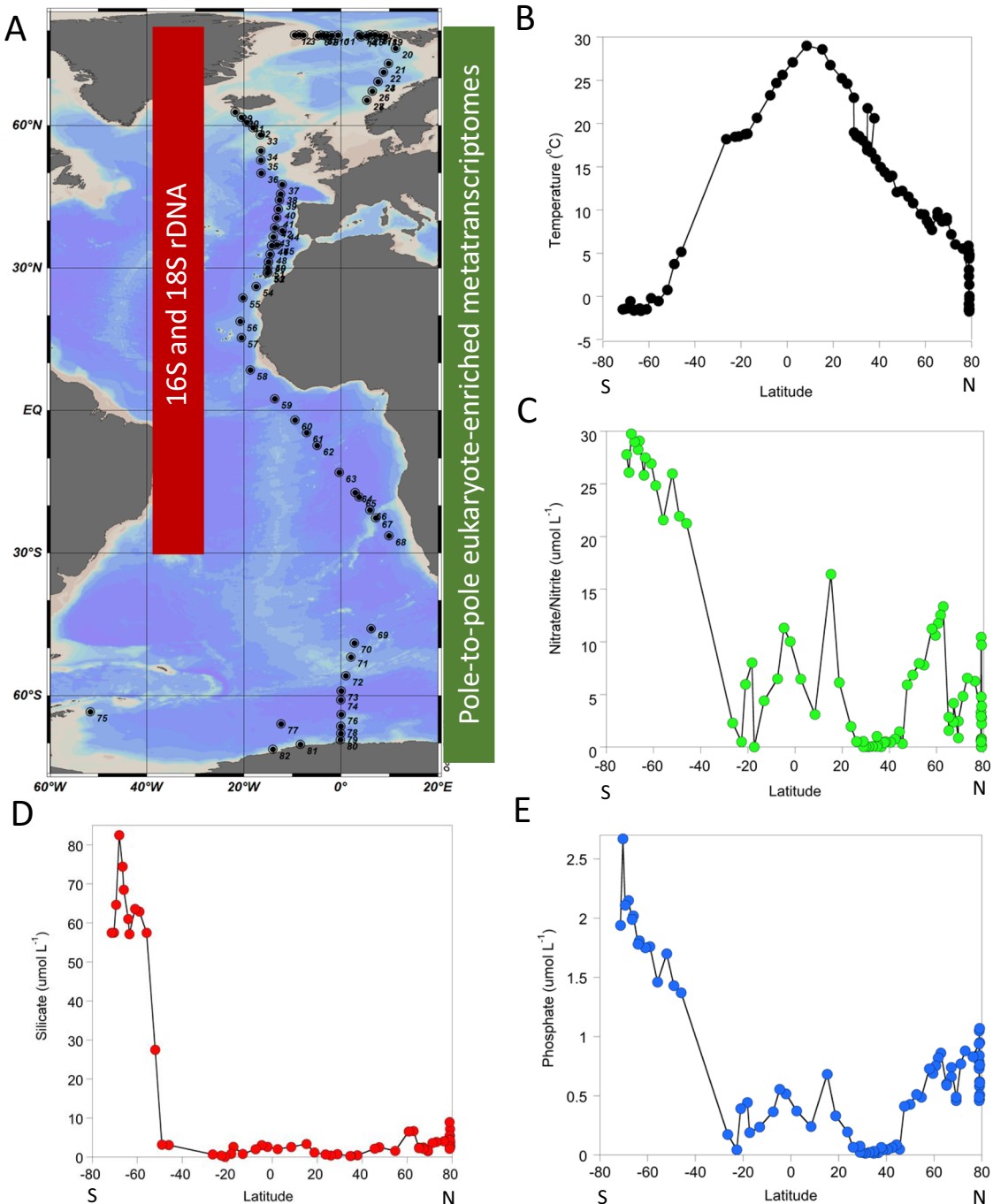

**Fig. 1 Sampling sites and environmental metadata. A** Stations for metatranscriptome sequencing (green) and 16 and 18S rDNA amplicon sequencing (red). Map was generated using Ocean Data view. **B** Latitude versus temperature (degree celsius). **C** Latitude versus nitrate and nitrite concentrations. **D** Latitude versus silicate concentrations. **E** Latitude versus phosphate concentrations. Nutrient concentrations in µmol L$^{-1}$.

—April to May 2011; ANT-XXIX/1 (PS81)—1st to 24th November 2012 and ANT-XXXII/2 (PS103)—20th December 2016 to 26th January 2017 and covers a transect of the Atlantic Ocean from Greenland to the Weddell Sea (71.36°S to 79.09°N).

The 79 eukaryotic metatranscriptomes were sequenced (Illumina HiSeq-2000 instrument) to an average depth of 251 Mbp each based on standard JGI protocols. These data were processed by the Integrated Microbial Genomes and Microbiomes (IMG) pipeline at JGI[27]. For estimating microbial diversity, 16S and 18S rDNA amplicon datasets were generated (Illumina MiSeq) with an average sequencing depth of 71.8 Mbp and 52.5 Mbp per sample, comprising an average of 393,247 and 142,693 sequences

per sample, respectively. A custom bioinformatics pipeline was built for 18S rDNA classifications including a model to normalise the copy number of 18S rDNAs according to the estimated genome sizes of diverse eukaryotic microbes (Supplementary Figs. 1, 2). Rarefaction analysis of all sequence datasets indicated that adequate sampling was achieved for all three types of datasets (Supplementary Fig. 3). Of the total number of contigs (34,241,890) in our metatranscriptome dataset, 36,354,419 non-redundant genes could be predicted, and from these genes ca. 31% (11,205,641 genes) could be assigned to a Pfam domain[28]. Most of the identified prokaryotic and eukaryotic taxa were present at more than 20 stations and had an evenness of J' ≥0.5

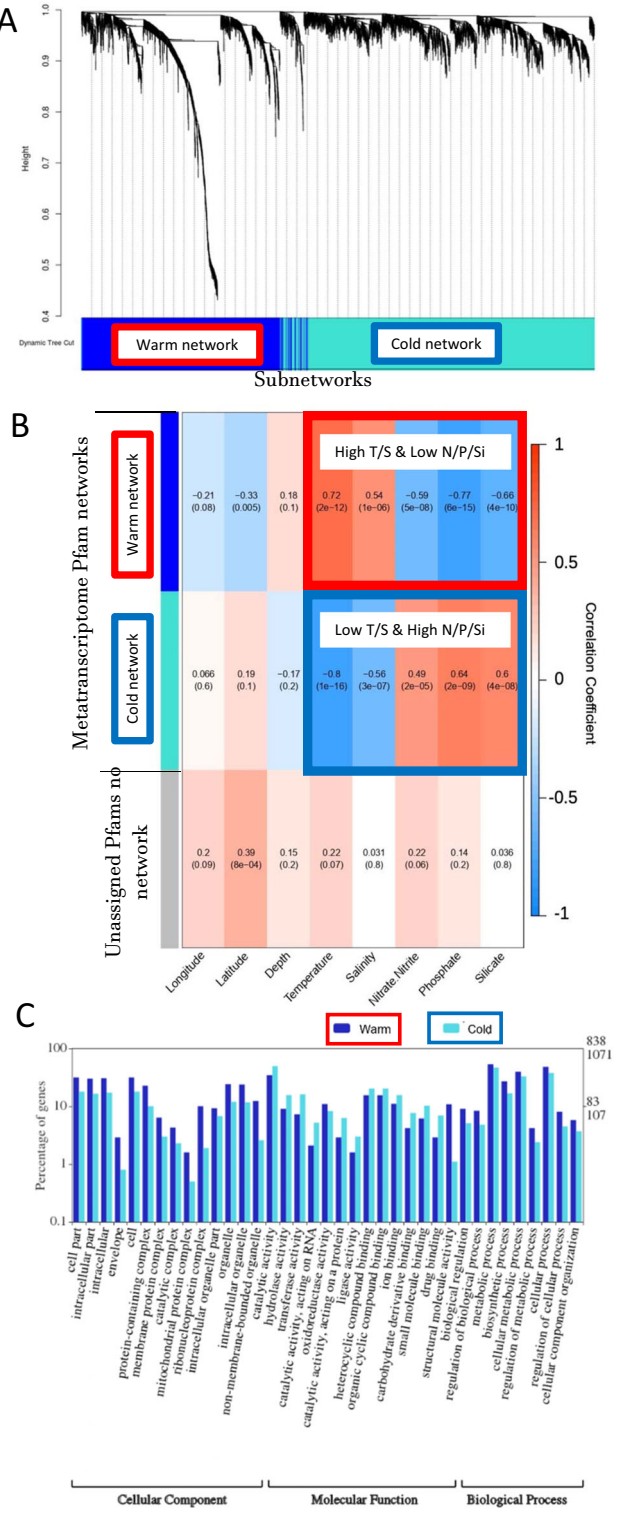

**Fig. 2 Co-occurrence networks of protein families in eukaryotic metatranscriptomes and their gene ontology.** On the log10-scaled gene counts of protein families (Pfams), two networks were found: **A** blue = warm (*n* = 1614) and turquoise = cold (*n* = 2369). **B** Co-occurrence analysis of Pfam protein families dataset, two networks were found, a turquoise (cold) and blue (warm), and also a grey (2 Pfams: no network). Correlation heatmap between the networks and environmental parameters. The colours correspond to the correlation values, red is positively correlated and blue is negatively correlated. The values in each of the squares correspond to the assigned Pearson correlation coefficient value on top and *p*-value in brackets below. **C** Gene ontology (GO) analysis of the co-occurrence of Pfam protein families dataset for both co-occurrence networks.

sampling (Fig. 1B–E; Supplementary Table 1). Temperatures in both hemispheres ranged from ca. −1.74 to 29.02 °C reflecting the pole to equator distribution of annual average upper ocean temperatures (Fig. 1B). Salinity varied between 31.0 and 36.9 PSU. Dissolved inorganic nutrients (µmol L⁻¹) were most highly concentrated in the Southern Ocean with minima for all nutrients at ca. 30°S/N (Fig. 1C–E). Based on a canonical correspondence analysis (CCA) all Pfams from metatranscriptomes against these individual environmental variables (Supplementary Fig. 6a, b), temperature was determined to account for the highest percentage of variation compared to all other environmental variables in each dataset. Temperature also had a significantly positive correlation ($R^2 \geq 0.63$; *p*-value $\leq 0.001$) with prokaryotic and eukaryotic diversity (Shannon Index) (Supplementary Fig. 7).

**Co-occurrence networks of expressed genes and microbial taxa.** The first pole-to-pole eukaryotic metatranscriptomes from chlorophyll *a* maximum layers (Fig. 1A) enabled us to provide insights into how global-scale environmental conditions in the upper ocean drive biogeographic differentiation of eukaryotic community gene expression. To identify which environmental variable was most responsible for a possible latitudinal differentiation in gene co-expression networks, we applied a weighted gene co-occurrence network analysis (WGCNA)[29] based on Pfam gene counts. Our WGCNA revealed that there were two gene co-expression networks only based on positive links (Fig. 2A, Supplementary Table 5). A correlation statistical analysis which is part of the WGCNA package was conducted. This involved taking each network's 'eigengene', a term used by WGCNA, which is the first principal component of a network, to be representative of that network in order to conduct a correlation analysis of networks to the environmental variables as shown in Fig. 2B. Based on this work, temperature was identified as the primary driver for both networks, which corroborates results from our CCA analysis (See above and Supplementary Fig. 6). Whereas salinity was co-correlated with temperature, the major inorganic nutrients such as nitrate, phosphate and silicate were significantly (*p*-value ≤0.001) anti-correlated to temperature and salinity. The gene co-expression network designated as blue (N = 1614 Pfams) has a strong positive relationship with temperature (correlation coefficient of +0.72; *p*-value = 2e−12), hence, this is considered to be the warm network. The network designated as turquoise (N = 2369 Pfams) has a strong negative relationship with temperature (correlation coefficient of −0.8; *p*-value = 1e−16), hence, this is considered to be the cold network. 7,172,786 genes with an average length of 757 bps were part of the cold network whereas the warm network was composed of 4,954,085 genes that had an average length of 655 bps. The average GC content of transcripts in the cold network was 51% and in the warm network was 52%. 831,540,849 reads of the cold network and

(Supplementary Figs. 4, 5). Only 22% of the 18S dataset could be assigned to taxa at the levels of species (Supplementary Figs. 4a, 6c), while for the 16S dataset, 47% could be assigned to taxa at the levels of genus (Supplementary Fig. 4b, Supplementary Fig. 6d). The metatranscriptomes represent a set of 36,354,419 non-redundant genes of which nearly 28% could be annotated as being of eukaryotic origin, and 31% had homology to known protein domains in the Pfam database. All sequence data were accompanied by measurements of temperature, salinity, dissolved inorganic nitrate/nitrite, phosphate and silicate at the depth of

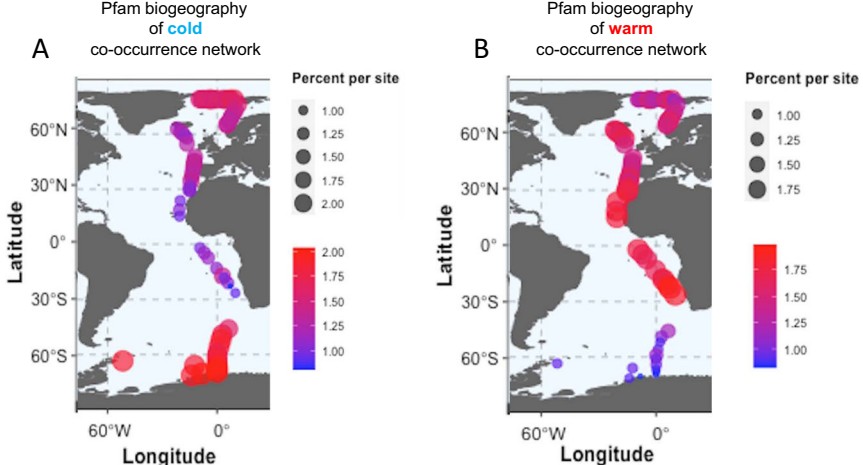

**Fig. 3 Biogeographical mapping of the node-specific abundance for each protein family (Pfams) network across all stations from pole to pole.** Contribution of Pfam containing sequences from individual metatranscriptome sites to corresponding protein family co-occurrence networks. Bubbles scaled according to percentage contribution to total abundance pool. **A** Pfam biogeography of cold co-occurrence network and **B** Pfam biogeography of warm co-occurrence network. Abundance is given in percentage contribution to the total sequence pool per site with increasing contribution from small to large circles and from blue to red.

1,239,584,159 reads of the warm network could be assigned Pfam domains. Unassigned Pfams designated as grey ($N = 2$ Pfams) did not form a co-expression network and had only a significantly positive correlation ($+0.39$; $p$-value $= 8e-04$) with latitude.

Gene ontology (GO) analyses with Pfams from both networks (Fig. 2C; Supplementary Fig. 8) showed that the cold network was enriched in several molecular functions associated with catalytic activity in general and specifically with acting on proteins and RNAs. Strongly enriched in the warm network were cellular components including mitochondria, ribosomes, non-membrane bound organelles, and the envelope.

The mapping of the node-specific Pfam abundance for each network across all stations is shown in Fig. 3A, B. Pfams of the cold network mainly recruit from the Southern Ocean and the Arctic (86.7% total) with the lowest abundance of Pfams mapping to stations between 30°N/S (13.3% total). In contrast, Pfams from the warm network were mainly recruited from the tropical and temperate North Atlantic (48.1% total). Interestingly, slightly more Pfams were recruited from the Arctic (38.7% total) then the Southern Ocean (13.2% total) for this network.

To reveal how environmental gradients from the Arctic to the equator influence associations between microbial eukaryotes and prokaryotes, we applied the same WGCNA[29] analysis as applied for the eukaryotic metatranscriptomes on log10 transformed normalized (according to genome size, Supplementary Fig. 2) abundances of 18S and 16S rDNA sequences. Co-occurrences were estimated on the normalized abundance of sequences at the species level for eukaryotes (18S) and genus level for prokaryotes (16S). Similar to the gene expression co-occurrence analysis, we obtained two major networks between eukaryotes and prokaryotes that correlated most strongly with temperature and latitude (Fig. 4A, B). Thus, similar to the gene co-expression networks, we identified a cold (Blue; $n = 51$ species; correlation coefficient of $\leq 0.79$; $p$-value $\leq 1e-10$) and a warm network (Turquoise; $n = 70$ species; correlation coefficient of $\geq 0.83$; $p$-value $\leq 3e-12$) of co-occurring eukaryotic and prokaryotic microbes (Supplementary Table 2). Unlike for the metatranscriptomes, there were no unassigned 16 and 18S sequences. In the cold network, green algae of the group Prasinophytes were species rich and the Prymnesiophyte *Phaeocystis cordata* had the highest number of connections to other species in this cluster (Supplementary Table 2). The prokaryotic community had several highly

connected bacterial taxa known to include cold-adapted species some of which co-occurring with diatoms (e.g. Glaciecola)[30]. Two bacterial taxa in this cluster (Herbaspirillum, Bradyrhizobium) are known to have species that have the ability to fix atmospheric $N_2$[31,32]. Although Coscinodiscophyceae were particularly abundant in cold waters of the Arctic, only one species (*Actinocyclus actinochilus*) was part of this cluster. The network from warm waters was very different in terms of species composition and co-occurrence patterns. Unlike in the cold network, cyanobacteria were among the most highly connected taxa including Prochlorococcus and Synechococcus. Small and mostly flagellated species from the group of Heterokontophyta dominated the most diverse group of eukaryotes in this cluster. There were also Dinoflagellates, Haptophytes and Pelagophytes. Many highly connected heterotrophic bacteria in this cluster are known to be associated with particles (e.g. soil, biofilm) and two taxa are known to have photoheterotrophic species that contain bacteriochlorophyll (Erythrobacter, Roseivivax)[33]. This cluster contained neither diatoms nor prasinophytes. There were eight shared classes of species in both co-occurrence networks namely Gammaproteobacteria, Alphaproteobacteria and Flavobacteriia. A full list of the classes of species can be found in Supplementary Table 2.

Biogeographical mapping of the node-specific 16 and 18S abundance for each network across all stations are shown in Fig. 4C, D. This revealed that 90.01% of sequences from the cold network were recruited from north of 60° in the Arctic Ocean with the opposite biogeographical recruitment pattern for the warm network (78.25 % from stations <60°N).

**The latitudinal differentiation (beta diversity) for expressed eukaryotic genes and microbial taxa.** As the co-occurrence analysis revealed for both expressed genes and taxa, that the environmental difference between polar and non-polar upper ocean waters appears to be most responsible for the geographical separation of algal microbiomes, we tested this result by calculating the ratio between regional and local sequence diversity (beta diversity) across all stations, which provides a measure of genetic differentiation between communities across latitudes. The partitioning of cold and warm co-occurrence networks suggests that there are major breakpoints in the genetic differentiation demarking the transition between polar and non-polar upper

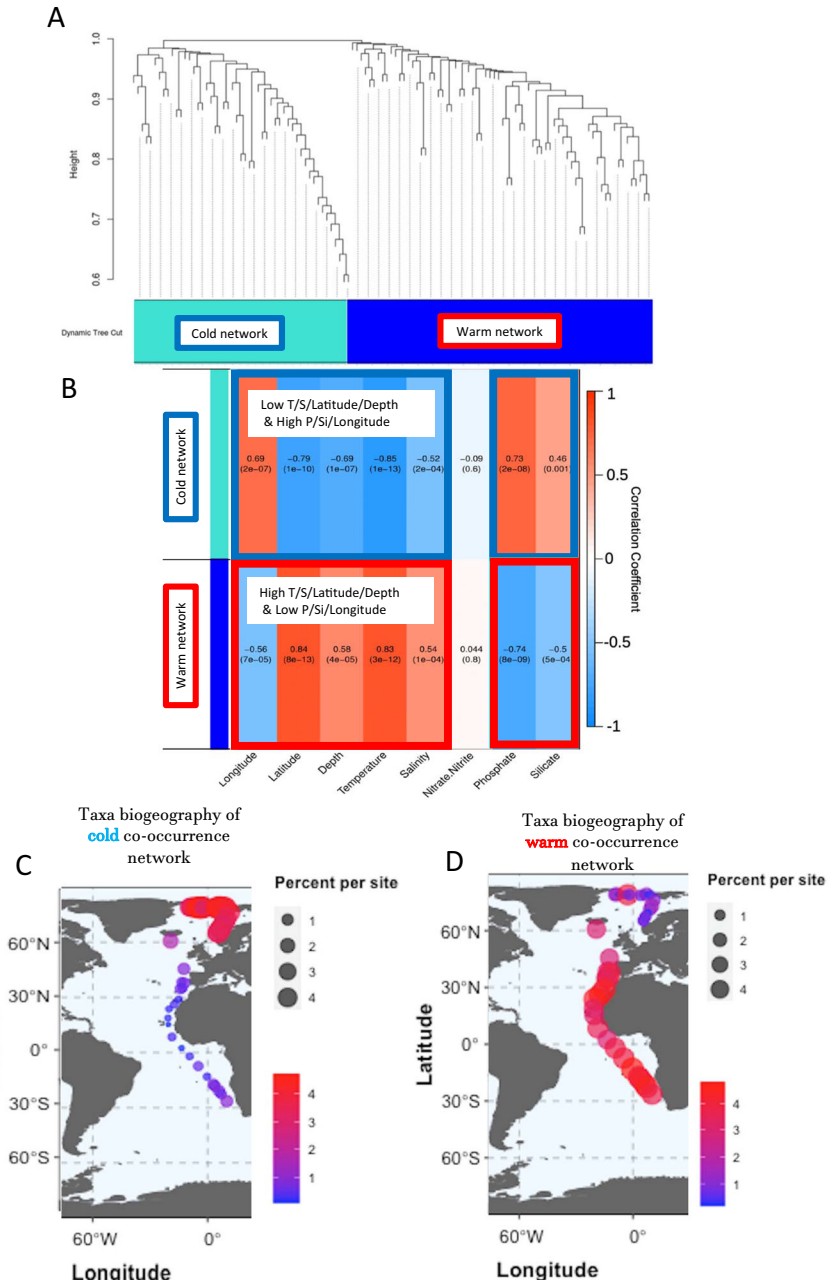

**Fig. 4 Co-occurrence networks of 16 and 18S rDNAs, their biodiversity and biogeographical mapping of the node-specific abundance for each taxonomic network across all stations from pole to pole.** On the log10 transformed abundances of 18S rDNA species level and 16S rDNA genus level, two networks were found: **A** cold ($n = 51$) and warm ($n = 70$). A list of species names and class names of the species can be found in the Supplementary Table 2. **B** Co-occurrence analysis of 18S rDNA species level and 16S rDNA genus level, two networks were found, a turquoise (cold) and blue (warm). Correlation heatmap between the networks and environmental parameters. The colours correspond to the correlation values, red is positively correlated and blue is negatively correlated. The values in each of the squares correspond to the assigned Pearson correlation coefficient value on top and p-value in brackets below. **C** Taxa biogeography of cold 16/18S co-occurrence network. **D** Taxa biogeography of warm 16/18S co-occurrence network. Abundance is given in percentage contribution to the total sequence pool per site with increasing contribution from small to large circles and from blue to red.

ocean ecosystems, with temperature and latitude likely being major drivers.

In order to test this hypothesis, we calculated a presence–absence matrix for each dataset. A multiple-site dissimilarity was performed on the presence–absence matrix with beta.pair, a function from the betapart R package and a dissimilarity index set by Sørensen[34]. These values were then plotted against all environmental variables, to enable us to get a range of values in which the breakpoint might be located. We then searched through these possible breakpoints for the one with

the lowest mean squared error. The search for breakpoints was performed using all environmental variables including nutrients and salinity as they are known to have an impact on microbial diversity and activity (Supplementary Figs. 9, 10)[14] Latitude correlates like temperature (Figs. 2B, 4B, 5A, B). Only the strong latitudinal gradient of temperature showed significant breakpoints in beta diversity, which largely separated cold from warm microbial communities and their associated metabolism (Fig. 5A). For metatranscriptomes, the breakpoint was estimated to be at 18.06 °C (Fig. 5A), for 16S we identified a breakpoint at ca.

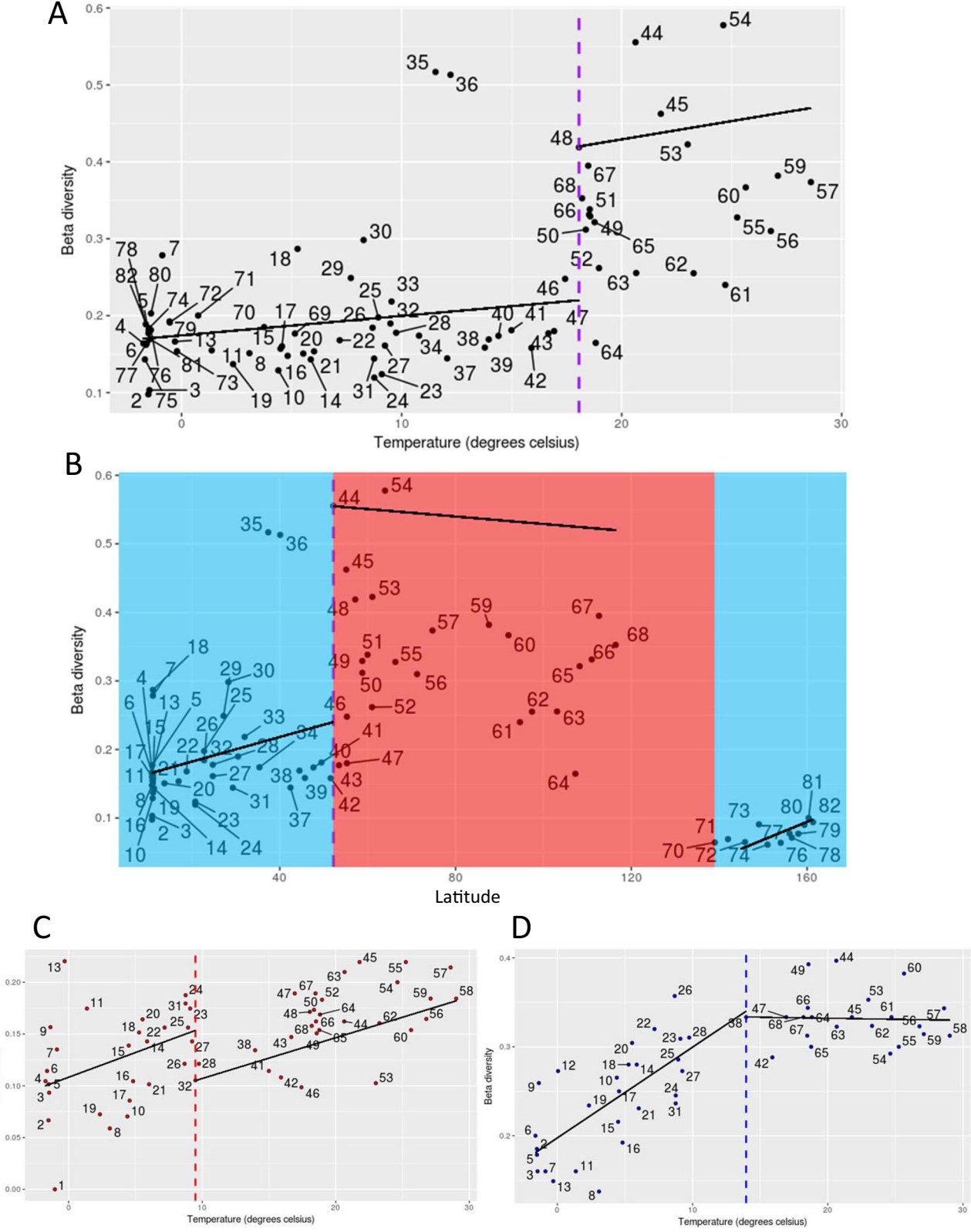

**Fig. 5 Beta diversity break-point analyses. A, B** Represent breakpoints of protein families as part of the metatranscriptome dataset. **C, D** Represent breakpoints of the 18S rDNA and 16S rDNA datasets. The numbers correspond to sample locations as shown in Fig. 1A. The y-axis represents beta diversity across all stations. The x-axis in **A**, **C** and **D** represents temperature and in **B** represents latitude. The horizontal lines indicate the breakpoints in beta diversity. For the Pfam protein families dataset in (**A**), the breakpoint is at 18.06 °C with a p-value of 3.741e−10. In **B** the breakpoint is at 52.167 degrees altered latitude (37.833 degrees latitude) with a p-value of 2.225e−07. For the 16S rDNA dataset in (**C**), the breakpoint is at 9.49 °C with a p-value of 1.413e−4. For the 18S rDNA dataset in (**D**), the breakpoint is at 13.96 °C with a p-value of 8.407e−11.

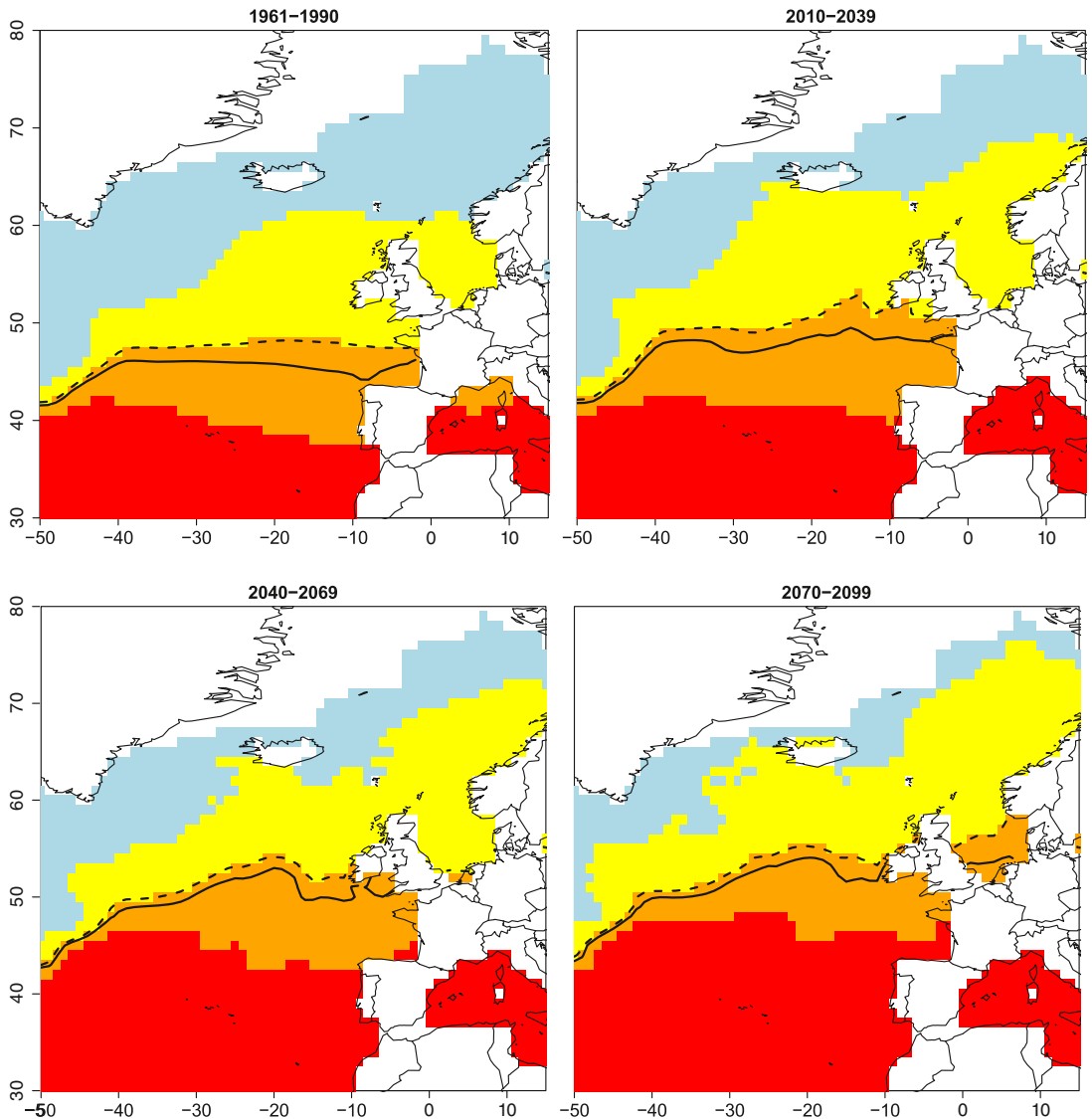

**Fig. 6 IPCC-based modelling of climate driven shifts in beta diversity breakpoints.** Observations (1961–1990) and modelled (2010–2099) changes over the 21st century, in the thresholds for breakpoints in beta diversity. Regions are shown as red for metatranscriptomes (>18.06 °C), orange for 18S (<18.06 °C, >13.96 °C), yellow for 16S (<13.96 °C, >9.49 °C) and blue for temperatures <9.49 °C for a 1961–1990 observations from the HadISST dataset. Modelled estimates temperatures from the HadGEM2-ES CMIP5 run for the 30-year averages, 2010–2039, 2040–2069, and 2070–2099, respectively. Temperatures from HadGEM2-ES have been calibrated to the HadISST observations as described in methods. Black solid line represents the 15 °C and the dashed line the 14 °C average upper ocean temperature.

9.49 °C (Fig. 5C) and 18S at 13.96 °C (Fig. 5D). The average temperature for the taxonomic and functional beta diversity of eukaryotic phytoplankton and their co-occurring bacteria is 14 °C ± 4.3. The metatranscriptome data enabled us to identify the geographical locations of the breakpoints as the dataset is pole to pole (Fig. 5B). The two breakpoints identified largely separate polar from non-polar oceans (Fig. 5B).

**Projection of geographical shifts in beta-diversity breakpoints across the North Atlantic.** The global ocean is a significant sink of heat with the consequence that the upper ocean has become warmer over the past 100 years due to the anthropogenic production of carbon dioxide. Thus, stratified warm-water masses expand pole-wards. This is of particular relevance in the North Atlantic and North Pacific and even the Arctic Ocean[35,36]. To simulate how warming of the North Atlantic might impact the beta-diversity breakpoints and therefore local changes in the algal

microbiomes, we utilised a model from the IPCC 5th Assessment Report. For estimates of changes over the 21st century, we use the RCP 8.5 HadGEM2-ES CMIP5 experiment[37]. A historical HadGEM2-ES experiment was also run for CMIP5, which we used to bias-correct the projected temperatures. The resulting shifts in breakpoints from these temperatures are shown in Fig. 6. Grid boxes that contain sea ice in the climatology were ignored from this analysis. Projections from the model show that the most affected geographical region in terms of shifts in the diversity of algal microbiomes over the coming decades is the area between 40 and 60° N, which includes the North Sea and most of the British Isles (Fig. 6).

**Discussion**
Our study has provided evidence that differences in environmental conditions between polar and non-polar upper oceans can explain the partitioning of co-occurring sequences into two major

algal microbiomes (Figs. 2–4). The latitudinal differentiation of their individual sequences based on beta diversity is mainly correlated with the latitudinal gradient of temperature in the upper ocean, especially at transition zones (breakpoints) between polar and non-polar oceans (Fig. 5), hence corroborating our WGCNA analysis (Figs. 2–4). However, many other environmental parameters including essential nutrients were either significantly negatively or positively correlated with temperature and latitude, suggesting that they also play an important role in the biogeographic differentiation of algal microbiomes in the upper ocean. The negative correlation of inorganic nutrients with temperature (Figs. 2B, 4B) reflects the observation that cold upper waters are usually nutrient-rich whilst warmer upper ocean waters tend to be nutrient poor considering global and annual averages[38]. Thus, differences in the physical structure (e.g. seasonally mixed vs permanently stratified water) of the upper ocean caused by latitudinal gradients of temperature might be the main reason for the separation into largely polar (cold) and non-polar (warm) algal microbiomes. The difference in recruiting sequences from polar vs non-polar oceans is larger for the two taxonomic networks (Fig. 4C, D) compared to the gene expression networks (Fig. 3A, B). Considering that the number and redundancy of expressed genes and Pfams in metatranscriptomes is significantly higher than the more distinct datasets of 16 and 18S sequences, this numerical difference may have contributed to differences in the degree of latitudinal partitioning. A reason for the stronger recruitment of Pfams from the Arctic (38.7% total) compared to the Southern Ocean (13.2% total) for the warm network might be due to the North Atlantic Current (NAC), which was sampled (Fig. 1), and likely carried microbes from lower latitudes as the NAC is a northward prolongation of the Gulf Stream. In contrast, the frontal system in the Southern Ocean represents a boundary system less prone to a poleward range shift of microbial species from lower latitudes[10]. Hence, a lower number of Southern Ocean Pfams were recruited for the warm co-occurrence network.

Although several global-scale studies, with Tara Oceans[22] being the most significant, have already revealed that temperature can be considered the best predictor of local epipelagic plankton diversity[9] our study has extended this work by including both polar oceans and by focusing on eukaryotic phytoplankton and their co-occurring prokaryotic microbes. Furthermore, this is the first study, at least to the best of our knowledge, which is based on latitudinal beta diversity to reveal genetic differentiation in marine microbial communities from pole to pole in relation to variable environmental conditions. Our results, therefore, provide insights into how changing environmental conditions correlate with biodiversity changes (breakpoints in beta diversity) subject to large-scale environmental fluctuation and disturbances[26]. This knowledge is essential for predicting the consequences of global warming (Fig. 6) and therefore may guide environmental management. Most previous studies compared local species diversity (alpha diversity) across latitudes[9]. Nevertheless, temperature was also identified as one of the most important variables explaining differences in species composition of local communities across large-scale latitudinal gradients.

The concept of ocean biogeochemical provinces (Longhurst provinces)[39] often matches local differences in upper-ocean microbiomes[14] and their linked biogeochemical activity such as nutrient and carbon cycling[40]. Although our study confirms the large-scale genetic differentiation of algal microbiomes between polar (ICE, SPSS) and non-polar Longhurst provinces (e.g. STSS, NHSTPS, SHSTPS) covered by our pole-to-pole transect, we did not identify geographic differentiation between any of the non-polar Longhurst provinces. Arguably, there are no stronger environmental differences than between polar and non-polar

upper oceans mainly caused by strong seasonality closer to the poles, overall low temperatures, the presence of sea ice, and differences in seasonal mixing[38]. Thus, environmental differences between polar and non-polar oceans may impose much stronger geographic differentiation in biodiversity of algal microbiomes and their expressed genes compared to environmental differences between Longhurst provinces of non-polar oceans (e.g. STSS, NHSTPS, SHSTPS). As the Arctic and the Southern Ocean do not significantly differ in their overall environmental conditions, this may explain why we have not seen a differentiation of algal microbiomes between both polar oceans. Hence, Pfams for the cold co-occurrence cluster have been recruited from both polar oceans (Fig. 3). The enrichment of GO terms for catalytic activity in the cold Pfam network likely reflects metabolic requirements to thrive under polar conditions. Most cold-adapted microbes optimise their enzymes to increase their catalytical activity at lower temperatures[41]. The optimization of enzymes to low temperature activity is usually facilitated by destabilisation of the molecular structures (e.g. active site). The enrichment of GO terms specifically for the catalytic activity of proteins and RNAs (Fig. 2C) suggests that these polar microbial communities have not only increased their catalytic activity of enzymes but also catalytic activity that acts to modify RNAs[42]. The GO enrichment of cellular components in the warm network (Fig. 2C) might reflect an increased turnover of subcellular compartments including their membranes due to increased metabolic activity (respiration in mitochondria) and stress (radical oxygen species) at higher temperatures, which is known to occur in microalgae[43].

The taxonomic differences based on 16 and 18S rDNA sequencing between cold and warm co-occurrence networks largely confirm differences in the biogeographical distribution of individual species across latitudinal regions of the global upper ocean[9,22,44–47]. For instance, Prochlorococcus and Synechoccus mainly dominate tropical and subtropical upper oceans together with eukaryotic pico- and nanoflagellates. Those taxa were found to be dominant in the warm network with a significant number of connections to additional taxa. In contrast, the cold network was characterised by abundant and well-connected sequences from phylogenetic groups known to include cold-adapted bacteria (e.g. Polaribacter, Glaciecola) and microalgae such as diatoms (e.g. *Actinocyclus actinochilus*) and Prymnesiophytes (e.g. *Phaecystis cordata*).

Interestingly, two previous studies have suggested a similar geographic partitioning but for phytoplankton productivity and mainly prokaryotic biodiversity. Behrenfeld et al.[38] identified that the physical environment of the upper ocean impacts the net primary production (NPP) of phytoplankton communities. On a global scale including polar oceans, they identified that differences in upper-ocean temperature and stratification across a latitudinal gradient were mainly responsible for the partitioning of NPP. The latter being higher in cold, nutrient-rich, and high-latitude regions whereas lower NPP was observed in warm, nutrient-poor and permanently stratified upper oceans. The demarcation zone between both global regions for NPP was estimated to be at approximately 15 °C on an annual average. This temperature is in good agreement with the average temperature for breakpoints in the taxonomic and functional beta diversity of eukaryotic phytoplankton and their co-occurring bacteria at 14 °C ± 4.3. A similar demarcation boundary was found for the latitudinal partitioning in diversity and activity of prokaryote-enriched metagenomes and metatranscriptomes, respectively[48]. Thus, our data together with these previous studied provide support for the hypothesis that environmental conditions separating cold (nutrient rich) from warm (nutrient poor) upper oceans are likely responsible for the latitudinal differentiation of algal microbiomes underpinning differences in ocean productivity and global biogeochemical cycles.

The latitudinal gradient of temperature caused by seasonal differences in solar radiation together with associated conditions such as differences in upper-ocean stratification and nutrient concentrations appear to be the main drivers. As the anthropogenic production of carbon dioxide raises global temperatures, which has already caused significant ocean warming, it is likely that the spatial distribution of algal microbiomes will change according to poleward shifts in geographical demarcation boundaries matching breakpoints in beta diversity of species and their gene pool. Our model for the North Atlantic shows that the area between 40 and 60° N might be affected the most over the next approximate 100 years as we forecast a complete replacement of cold algal microbiomes (Fig. 6) in this geographical area. As the area between 40 and 60° N is known to be nutrient rich and, therefore, productive especially the North Sea, a replacement of current microbial communities is likely to have significant impact on food webs including fisheries with consequences for associated industries.

Taken together, our study confirms the latitudinal distribution pattern in local (alpha) diversity of complex marine microbial communities with a significant decrease from the equator towards polar ecosystems (Supplementary Fig. 7)[9]. However, pole-to-pole datasets, which represent a more complete spectrum of environmental variables, offer the opportunity to identify the most pronounced differences in the variation of alpha diversity across larger biogeographic regions (beta diversity). The latter, to the best of our knowledge, has never been estimated before for oceanic microbes although this knowledge is instrumental for spatial scaling of changes in diversity, i.e. loss and gain[26]. The application of beta diversity to pole-to-pole algal microbiomes revealed for the first time that physico-chemical differences between polar and non-polar upper oceans have a strong influence not only on changes in their diversity but also the gene expression activity of their primary producers. Consequently, there appear to be ecological boundaries in sub-polar oceans of both hemispheres, which not only alter the spatial scaling of algal microbiomes (breakpoints in beta diversity), but also shift polewards due to global warming.

## Methods

**Research cruises.** This dataset consists of sequence data from 4 separate cruises: ARK-XXVII/1 (PS80)—17th June to 9th July 2012; Stratiphyt-II— April to May 2011; ANT-XXIX/1 (PS81)—1st to 24th November 2012 and ANT-XXXII/2 (PS103)—16th December 2016 to 3rd February 2017 and covers a transect of the Atlantic Ocean from Greenland to the Weddell Sea (71.36°S to 79.09°N) (Supplementary Table 1). In order to study the composition, distribution and activity of microbial communities in the upper ocean across the broadest latitudinal ranges possible, samples have been collected during four field campaigns as shown in Fig. 1A. The first collection of samples was collected in the North Atlantic Ocean from April to May 2011 by Dr. Willem van de Poll of the University of Groningen, Netherlands and Dr. Klaas Timmermans of the Royal Netherlands Institute for Sea Research. The second set of samples was collected in the Arctic Ocean from June to July 2012, and the third set of samples was collected in the South Atlantic Ocean from October to November 2012. Both of which were collected by Dr. Katrin Schmidt of the University of East Anglia. The final set of samples was collected in the Antarctic Ocean from December 2016 to January 2017 by Dr. Allison Fong of the Alfred-Wegener-Institute for Polar and Marine Research, Bremerhaven, Germany.

**Sampling.** Water samples from the Arctic Ocean and South Atlantic Ocean expeditions were collected using 12 L Niskin bottles (Rosette sampler with an attached Sonde (CTD, conductivity, temperature, depth) either at the chlorophyll maximum (10–110 m) and/or upper of the ocean (0–10 m). As soon as the rosette sampler was back on board, water samples were immediately transferred into plastic containers and transported to the laboratory. All samples were accompanied by measurements on salinity, temperature, sampling depth and silicate, nitrate, phosphate concentration (Supplementary Table 1). Water samples were pre-filtered with a 100 μm mesh to remove larger organisms and subsequently filtered onto 1.2 μm polycarbonate filters (Isopore membrane, Millipore, MA, USA). All filters were snap frozen in liquid nitrogen and stored at −80 °C until further analysis.

Water samples from the North Atlantic Ocean cruise were also taken with 12 L Niskin bottles attached to a Rosette sampler with a Sonde. However, these samples were filtered onto 0.2 μm polycarbonate filters (Isopore membrane, Millipore, MA, USA) without pre-filtration but snap frozen in liquid nitrogen and stored at −80 °C as the other samples.

Water samples from the Southern Ocean cruise were taken with 12 L Niskin bottles attached to an SBE911plus CTD system equipped with 24 Niskin samplers. These samples were filtered onto 1.2 μm polycarbonate membrane filters (Merck Millipore, Germany) in a container cooled to 4 °C and snap frozen in liquid nitrogen and stored at −80 °C as the other samples. Environmental data recorded at the time of sampling can be found in Supplementary Table 1.

**DNA extractions: Arctic Ocean and South Atlantic Ocean samples.** DNA was extracted with the EasyDNA Kit (Invitrogen, Carlsbad, CA, USA) with modification to optimise DNA quantity and quality. Briefly, cells were washed off the filter with pre-heated (65 °C) Solution A and the supernatant was transferred into a new tube with one small spoon of glass beads (425–600 μm, acid washed) (Sigma-Aldrich, St. Louis, MO, USA). Samples were vortexed three times in intervals of 3 s to break the cells. RNase A was added to the samples and incubated for 30 min at 65 °C. The supernatant was transferred into a new tube and Solution B was added followed by a chloroform phase separation and an ethanol precipitation step. DNA was pelleted by centrifugation and washed several times with isopropanol, air dried and suspended in 100 μL TE buffer (10 mM Tris-HCl, pH 7.5, 1 mM EDTA, pH 8.0). Samples were snap frozen in liquid nitrogen and stored at −80 °C until sequencing.

**DNA extractions: North Atlantic Ocean samples.** North Atlantic Ocean samples were extracted with the ZR-Duet™DNA/RNA MiniPrep kit (Zymo Research, Irvine, USA) allowing simultaneous extraction of DNA and RNA from one sample filter. Briefly, cells were washed from the filters with DNA/RNA Lysis Buffer and one spoon of glass beads (425–600 μm, Sigma-Aldrich, MO, USA) was added. Samples were vortexed quickly and loaded onto Zymno-Spin™IIIC columns. The columns were washed several times and DNA was eluted in 60 μmL, DNase-free water. Samples were snap frozen in liquid nitrogen and stored at −80 °C until sequencing.

**DNA extractions: Southern Ocean samples.** DNA from the Southern Ocean samples was extracted with the NucleoSpin Soil DNA extraction kit (Macherey-Nagel) following the manufacturer's instructions. Briefly, cells were washed from the filters with DNA Lysis Buffer and into a lysis tube containing glass beads was added. Samples were disrupted by bead beating for 2 × 30 s interrupted by 1 min cooling on ice and loaded onto the NucleoSpin columns. The columns were washed three times and DNA was eluted in 50 μL, DNase-free water. Samples were stored at −20 °C until further processing.

**Amplicon sequencing of 16S and 18S rDNA.** All extracted DNA samples were sequenced and pre-processed by the Joint Genome Institute (JGI) (Department of Energy, Berkeley, CA, USA). iTAG amplicon sequencing was performed at JGI with primers for the V4 region of the 16S (FW(515F): GTGCCAGCMGCCGCG GTAA; RV(806R): GGACTACNVGGGTWTCTAAT)[49] and 18S (FW(565F): CCAGCASCYGCGGTAATTCC; RV(948R): ACTTTCGTTCTTGATYRA)[50]. (Supplementary Table 6) rRNA gene (on an Illumina MiSeq instrument with a 2 × 300 base pairs (bp) read configuration[51]. 18S sequences were pre-processed, this consisted of scanning for contamination with the tool Duk (US Department of Energy Joint Genome Institute (JGI), 2017,a) and quality trimming of reads with cutadapt[52]. Paired end reads were merged using FLASH[53] with a max mismatch set to 0.3 and min overlap set to 20. A total of 54 18S samples passed quality control after sequencing. After read trimming, there was an average of 142,693 read pairs per 18S sample with an average length of 367 bp and 2.8 Gb of data over all samples.

16S sequences were pre-processed, this consisted of merging the overlapping read pairs using USEARCH's merge pairs[54] with the parameter minimum number of differences (merge max diff pct) set to 15.0 into unpaired consensus sequences. Any reads that could not be merged are discarded. JGI then applied the tool USEARCH's search oligodb tool with the parameters mean length (len mean) set to 292, length standard deviation (len stdev) set to 20, primer trimmed max difference (primer trim max diffs) set to 3, a list of primers and length filter max difference (len filter max diffs) set to 2.5 to ensure the Polymerase Chain Reaction (PCR) primers were located with the correct direction and inside the expected spacing. Reads that did not pass this quality control step were discarded. With a max expected error rate (max exp err rate) set to 0.02, JGI evaluated the quality score of the reads and those with too many expected errors were discarded. Any identical sequence was de-duplicated. These are then counted and sorted alphabetically for merging with other such files later. A total of 57 × 16S samples passed quality control after sequencing. There was an average 393,247 read pairs per sample and an average base length of 253 bp for each sequence with a total of 5.6 Gb.

**RNA extractions: Arctic Ocean and Atlantic samples.** RNA from the Arctic and Atlantic Ocean samples was extracted using the Direct-zol RNA Miniprep Kit (Zymo Research, USA). Briefly, cells were washed off the filters with Trizol into a tube with one spoon of glass beads (425–600 μm, Sigma-Aldrich, MO, USA). Filters

were removed and tubes bead beaten for 3 min. An equal volume of 95% ethanol was added, and the solution was transferred onto Zymo-Spin™ IICR Column and the manufacturer instructions were followed. Samples were treated with DNAse to remove DNA impurities, snap frozen in liquid nitrogen and stored at −80 °C until sequencing.

**RNA extractions: Southern Ocean**. RNA from the Southern Ocean samples was extracted using the QIAGEN RNeasy Plant Mini Kit (QIAGEN, Germany) following the manufacturer's instructions with on-column DNA digestion. Cells were broken by bead beating like for the DNA extractions before loading samples onto the columns. Elution was performed with 30 μm RNase-free water. Extracted samples were snap frozen in liquid nitrogen and stored at −80 °C until sequencing.

**Metatranscriptome sequencing**. All samples were sequenced and pre-processed by the U.S. Department of Energy Joint Genome Institute (JGI). Metatranscriptome sequencing was performed on an Illumina HiSeq-2000 instrument[27]. A total of 79 samples passed quality control after sequencing with 19.87 Gb of sequence read data over all samples for analysis. This comprised a total of 34,241,890 contigs, with an average length of 503 and an average GC% of 51%. This resulted in 36354419 of non-redundant genes detected.

JGI employed their suite of tools called BBTools[55] for preprocessing the sequences. First, the sequences were cleaned using Duk a tool in the BBTools suite that performs various data quality procedures such as quality trimming and filtering by kmer matching. In our dataset, Duk identified and removed adaptor sequences, and also quality trimmed the raw reads to a phred score of Q10. In Duk the parameters were; kmer-trim (ktrim) was set to r, kmer (k) was set to 25, shorter kmers (mink) set to 12, quality trimming (qtrim) was set to r, trimming phred (trimq) set to 10, average quality below (maq) set to 10, maximum Ns (maxns) set to 3, minimum read length (minlen) set to 50, the flag "tpe" was set to t, so both reads are trimmed to the same length and the "tbo" flag was set to t, so to trim adaptors based on pair overlap detection. The reads were further filtered to remove process artefacts also using Duk with the kmer (k) parameter set to 16.

BBMap[55] is another tool in the BBTools suite, that performs mapping of DNA and RNA reads to a database. BBMap aligns the reads by using a multi-kmer-seed-and-extend approach. To remove ribosomal RNA reads, the reads were aligned against a trimmed version of the SILVA database using BBMap with parameters set to; minratio (minid) set to 0.90, local alignment converter flag (local) set to t and fast flag (fast) set to t. Also, any human reads identified were removed using BBMap.

BBmerge[56] is a tool in the BBTools suite that performs the merging of overlapping paired end reads (Bushnell, 2017). For assembling the metatranscriptome, the reads were first merged with the tool BBmerge, and then BBNorm was used to normalise the coverage so as to generate a flat coverage distribution. This type of operation can speed up assembly and can even result in an improved assembly quality.

Rnnotator[52] was employed for assembling the metatranscriptome samples 1–68. Rnnotator assembles the transcripts by using a de novo assembly approach of RNA-Seq data and it accomplishes this without a reference genome[52]. MEGAHIT[57] was employed for assembling the metatranscriptome samples 69–82. The tool BBMap was used for reference mapping, the cleaned reads were mapped to metagenome/isolate reference(s) and the metatranscriptome assembly.

**Metatranscriptome analysis**. JGI performed the functional analysis on the metatranscriptomic dataset. JGI's annotation system is called the Metagenome Annotation Pipeline (MAP) (v4.15.2)[27]. JGI used HMMER 3.1b2[58] and the Pfam v30[59] database for the functional analysis of our metatranscriptomic dataset. This resulted in 11,205,641 genes assigned to one or more Pfam domain. This resulted in 8379 Pfam functional assignments and their gene counts across the 79 samples. The files were further normalised by applying hits per million.

**18S rDNA analysis**. A reference dataset of 18S rRNA gene sequences that represent algae taxa was compiled for the construction of the phylogenetic tree by retrieving sequences of algae and outgroups taxa from the SILVA database (SSUREF 115)[60] and Marine Microbial Eukaryote Transcriptome Sequencing Project (MMETSP) database[61]. The algae reference database consists of 1636 species from the following groups: Opisthokonta, Cryptophyta, Glaucocystophyceae, Rhizaria, Stramenopiles, Haptophyceae, Viridiplantae, Alveolata, Amoebozoa and Rhodophyta. A diagram of the 18S classification pipeline can be found in Supplementary Fig. 1. In order to construct the algae 18S reference database, we first retrieved all eukaryotic species from the SILVA database with a sequence length of > = 1500 base pairs (bp) and converted all base letters of U to T. Under each genus, we took the first species to represent that genus. Using a custom written script (https://github.com/SeaOfChange/SOC/blob/master/get_ref_seqs.pl), the species of interest (as stated above) were selected from the SILVA database, classified with NCBI taxa IDs and a sequence information file produced that describes each of the algae sequences by their sequence ID and NCBI species ID. Taxonomy from the NCBI database, eukaryote sequences from the SILVA database and a list of algal taxa including outgroups were used as input for the script. This information was combined with the MMETSP database excluding duplications.

The algae reference database was clustered to remove closely related sequences with CD-HIT (4.6.1)[62] using a similarity threshold of 97%. Using ClustalW (2.1)[63] we aligned the reference sequences with the addition of the parameter iteration numbers set to 5. The alignment was examined by colour coding each species to their groups and visualising in iTOL[64]. It was observed that a few species were misaligning to other groups and these were then deleted using Jalview[65]. The resulting alignment was tidied up with TrimAL (1.1)[66] by applying parameters to delete any positions in the alignment that have gaps in 10% or more of the sequence, except if this results in less than 60% of the sequence remaining. A maximum likelihood phylogenetic reference tree and statistics file based on our algae reference alignment was constructed by employing RaxML (8.0.20)[67] with a general time reversible model of nucleotide substitution along with the GAMMA model of rate heterogeneity. For a description of the lineages of all species back to the root in the algae reference database, the taxa IDs were submitted for each species to extract a subset of the NCBI taxonomy with the NCBI taxtastic tool (0.8.4)[68] Based on the algae reference multiple sequence alignment, with HMMER3 (3.1B1)[69] a Profile HMM was created. A placer reference package using taxtastic was generated, which produced an organized collection of all the files and taxonomic information into one directory. With the reference package, a SQLite database was created using pplacer's Reference Package PReparer (rppr). With hmmalign, the query sequences were aligned to the reference set and created a combined Stockholm format alignment. Pplacer (re-aligned to the reference set and created a combined Stockholm format alignment. Pplacer (1.1)[70] was used to place the query sequences on the phylogenetic reference tree by means of the reference alignment according to a maximum likelihood model[70] The place files were converted to CSV with pplacer's guppy tool; in order to easily take those with a maximum likelihood score of > = 0.5 and counted the number of reads assigned to each classification. This resulted in 6,053,291 reads that were taxonomically assigned taken for analysis.

**Normalisation of 18S rDNA gene copy number**. 18S rDNA gene copy number vary widely among eukaryotes. In order to create an estimate of abundances of the species in the samples the data had to be normalised. Previous work has explored the link between copy number and genome size[71]. However, there is not a single database of 18S rDNA gene copy numbers for eukaryote species. In order to address this, gene copy number and related genome sizes of 185 species across the eukaryote tree was investigated and plotted (Supplementary Fig. 2, Supplementary Table 4)[68,71–79]. Based on the log transformed data, a significant correlation with a R2 of 0.55 with a p-value < 2.2e−16 between genome size and 18S copy number was observed. A regression equation was determined (f(x) = 0.66X + 0.75) as shown in Supplementary Fig. 2.

To derive this equation, the genome sizes for the species in the reference datasets were retrieved from the NCBI genome database. Since some of the genome sizes were unavailable, for species with missing genome sizes, an average of available genome sizes in closely related species was taken instead. More specifically, first a taxonomic lineage of the relevant subset of the NCBI database was obtained by submitting the taxa IDs using the NCBI taxtastic tool[68]. Average genome sizes were then calculated by utilizing the parent ID and taxa ID columns and the known genome sizes of the lowest common ancestor. The 18S datasets were normalised by assigning their genome sizes using the regression equation. The files were further normalised by applying the hits per million reads method.

**18S rDNA file preparation**. In our 18S rDNA dataset, we had taxonomic assignments from the eukaryote node down to the species nodes. We employed Metagenome Analyzer (MEGAN) (5.10.3)[80] to cut out specific taxonomic levels. In MEGAN, we extracted the classifications at the taxonomic rank of species. This consisted of a file being generated for each station that contained the species names and their assigned abundances. The files were further normalised to hits per million.

In MEGAN, we extracted the leaves of the taxonomy tree at the rank of class and above but excluded assignments to the eukaryote node. Firstly, this consisted of a file being generated for each station that contained all assignments to the class nodes as well as any assignments under their respective lineages down to species being summed up under the individual class node. Secondly, we included nodes that were not highlighted for class taxonomic level on the leaves of the tree in MEGAN. These leaves were not highlighted because in NCBI taxonomy there are species that do not have a taxonomy designation at every taxonomy level. We took the nodes that were not highlighted on leaves of the tree and summed them together within their respective lineages and placed them under a new name. For example, under the phylum Rhizaria, on the leaves of the tree, there is Cercozoa, Gromiidae and unclassified Rhizaria which are not highlighted. Their abundance was summed together and renamed Nc. Rhizaria, "Nc." standing for "No class". The abundances assigned to Rhizaria were not included in this calculation. The leaves of the tree made up 34% of the total 18S rDNA dataset. The internal nodes between the leaves of the tree at the taxonomic rank of class and the eukaryote node was given a "U." in front of their name, "U." standing for "Unknown". This was done to highlight that while they are of course associated with the lower lineages they are in fact considered separate, as those assignments to those nodes could not be determined any lower. The internal nodes made up 29% of the total 18S rDNA dataset.

The abundance assigned to the eukaryote node was excluded from our analysis as these sequences could not be classified lower. This comprised of a total of 37% of the 18S rDNA dataset. A file was generated for each station that contained the class nodes, "Nc." nodes and "U." nodes with their respective abundances. The files were further normalised to hits per million. Throughout the paper we refer to the analysis of these files at the taxonomic rank of class.

**16S rDNA analysis**. JGI performed the classification analysis on the 16S rDNA dataset[81,82]. JGI's 16S rDNA classification pipeline (JGI pipeline iTagger v2.1 16S classification pipeline) consists of firstly removing samples with less than 1000 sequences. The remaining samples and the de-duplicated identical sequences from the preprocessing step are then combined and their sequences organized by decreasing abundance. The sequences are divided out based on the criterion as to whether they contained a cluster centroid with a minimum size of at least 3 copies. The low-abundance sequences are put aside and not used for clustering. USEARCH's[83] cluster otus command is employed to incrementally cluster the clusterable sequences. This begins at 99% identity and the radius is increased by 1% for each iteration until a OTU clustering identity of 97% is reached. At each step, the sequences are sorted by decreasing abundance. Once clustering is complete, USEARCH's usearch global is used to map the low-abundance sequences to the cluster centroids. These are added to OTU counts if they were in the prescribed percent identity threshold. If they do not fall within this prescribed percent identity threshold they are discarded. USEARCH's UTAX along with the SILVA database is used to evaluate the clustered centroid sequences. The predicted taxonomic classifications are then filtered with a cutoff of 0.5. Any chloroplast sequences identified are removed. The final accepted OTUs and read counts for each sample are finally placed in a taxonomic classification file.

**Normalisation of 16S rDNA gene copy number**. In order to normalise the 16S copy number, the 16S copy numbers for the species in the dataset were retrieved from the Ribosomal RNA Operon Copy Number Database (rrnDB)[84] The rrnDB database version 5.3 consisted at the time of 3021 bacterial entries. Firstly, since multiple entries of a species are in the rrnDB database due to the presence of different strains, we obtained an average copy number for each species in the rrnDB database, which resulted in 2876 species entries. The higher taxonomic levels for the rrnDB species needed to be established so that we could calculate their average copy number. For a description of the lineages of all species back to the root in the rrnDB database, we submitted the species names for each entry to extract a subset of the NCBI taxonomy with the NCBI taxtastic tool[68] thus producing a Taxtastic file. The Taxtastic file based on species from the rrnDB database was used to calculate the average copy number for higher taxonomic levels from the known copy number species level, with the assistance of the parent id and taxa id layout in the Taxtastic file. A Taxtastic file based on 16S rDNA species from our dataset was generated and we assigned our 16S species entries a copy number from species to root from the prepared average copy number rrnDB Taxtastic file. Not all copy numbers in the 16S rDNA dataset were known. We, therefore, took the average of closely related species from the above taxonomic level of those we could get and took that as the copy number for those that were missing in our dataset. The 16S dataset was normalised by dividing by the assigned copy number. The files were further normalised by applying the hits per million reads method.

**16S rDNA file preparation**. In our 16S rDNA dataset, we had taxonomic assignments from the bacteria node down to the genus nodes. We extracted the classifications at the taxonomic rank of genus. This consisted of a file being generated for each station that contained the genus names and their assigned abundances. The files were further normalised by applying the hits per million reads method.

We extracted the leaves of the tree that included class nodes and "Nc." nodes with their respective abundances. This step resulted in 94% of the 16S rDNA dataset. Also, we extracted the internal nodes and placed "U." in front of their names. This resulted in 3% of the 16S rDNA dataset. The abundance assigned to the bacteria node was excluded from our analysis and this comprised of a total of 3% of the 16S rDNA dataset. We generated a file for each station that contained the class nodes, "Nc." nodes and "U." nodes with their respective abundances. The files were further normalised by applying the hits per million reads method. Throughout the paper we refer to the analysis of these files at the taxonomic rank of class.

**Statistical analysis**. Alpha diversity (Shannon index) in relation to environmental covariates

The Shannon index H'[85] was used to calculate abundance weighted richness per station. The Shannon index was used over the Simpson index as the latter is heavily weighted towards the most abundant orders. The Shannon index was calculated based on the following equation:

$$H' = - \sum_{i=1}^{S} p_i \ln p_i$$

Environmental covariates were related to the Shannon index (H') by fitting generalized linear models. Step-by-step backwards selection of covariates was used

for model building, removing non-significant covariates until remaining covariates were significant at a p-value < 0.05.

Beta diversity in relation to environmental factors was calculated across the transect based on a Hellinger transformed class abundance matrix using the vegdist function of the vegan package[86]. The Bray-Curtis dissimilarity index[87] was used as a measure of beta-diversity and was calculated based on the following equation:

$$BC_{ij} = \sum \frac{|n_{ik} - n_{jk}|}{(n_{ik} + n_{jk})}$$

**Evenness and occupancy**. An abundance, station evenness and occupancy plots were produced for each 18S rDNA class level ($n = 54$) and 16S rRNA class level ($n = 57$) (Supplementary Fig. 5, Supplementary Table 3) The x-axis represents the number of times that class taxonomy occurs across the stations. The y-axis represents the evenness of that class taxonomy across stations it occurs in. This was calculated using a Dispersion index, which is a varient of J' Pielou's evenness[88] and based on H' of Shannon[85,89]. Each circle represents a class taxonomy abundance. The size of each circle is resized by replacing the area of the circle which represented the total abundance for that class with square root of the abundance divided by pi.

**Canonical correspondence analyses (CCAs)**. The R package VEGAN[90] was employed to perform a Canonical Correspondence Analysis (CCA) on each dataset of 18S, 16S and metatranscript Pfam against the individual environmental variables. The environmental data consisted of temperature, salinity, nitrate/nitrite, phosphate and silicate (Supplementary Fig. 6).

**Network analysis**. A network analysis was performed using the R package Weighted Gene Co-Expression Network Analysis (WGCNA)[91] The first analysis was performed on samples of combined prokaryotes at the taxonomic rank of genus and on eukaryotes at the taxonomic rank of species to describe networks derived from their log10-scaled abundances. The prokaryotes and eukaryotes normalised files were combined for each station. A signed adjacency measure for each lineage was determined by raising the absolute value of the Pearson correlation coefficient to the power of 11. A topological overlap measure (TOM) was calculated from the resulting adjacency matrix. Hierarchical clustering was carried out on the TOM measure, which resulted in two networks being discovered in the network (Fig. 4). The second analysis was performed on samples of the metatranscriptome Pfam dataset to describe networks derived from their log10-scaled gene counts. A signed adjacency measure for each lineage was determined by raising the absolute value of the Pearson correlation coefficient to the power of 12. A topological overlap measure (TOM) was calculated from the resulting adjacency matrix. Hierarchical clustering was carried out on the TOM measure, which resulted in two networks being discovered in the network (Fig. 2, Supplementary Table 5).

When incorporating environmental data, latitude values were redefined, so that the North pole is 0°, the Equator is 90° and the South pole is 180°. Unaltered environmental data can be found in Supplementary Table 1.

**Beta diversity break-point analysis**. The break-point analysis is based on the methodology from ref. [92]. The beta diversity indices used in the break-point analyses is the Sørensen indices. A breakpoint was determined and plotted for each of the Pfam protein families, 18S rDNA and 16S rDNA datasets. Breakpoints in the 18S and 16S rDNA datasets were investigated between the temperature range of 7 °C to 29.02 °C. When incorporating environmental data, latitude values were redefined, so that the North pole is 0°, the Equator is 90° and the South pole is 180°. Unaltered environmental data can be found in Supplementary Table 1.

The break-point analysis was generated using piecewise regression in R. This was calculated by firstly producing a presence–absence matrix for each dataset. A multiple-site dissimilarity was performed on the presence–absence matrix with beta.pair, a function from the betapart R package and a dissimilarity index set to Sørensen, thus produced a distance object called beta.sor[34]. Outliers were identified with bagplot, a function from the aplpack R package and then removed from the analyses. Remaining values were then plotted against the environmental variable (temperature or altered latitude), these were searched through for possible breakpoints, that is for the one with the lowest mean squared error.

For the 18S rDNA and 16S rDNA datasets, a number of samples in the North Atlantic Ocean did not pass quality control before sequencing. Due to this, when performing the 18S rDNA and 16S rDNA break-points analyses there were gaps in each of the datasets plots in the North Atlantic Ocean region. To investigate the effects of the missing samples, four model scenarios were produced to mimic the missing samples. The first model scenario involved filling in beta diversity values for the missing North Atlantic Ocean with current closest by latitude stations. This resulted in breakpoints for the 18S and 16S rDNA of 20.66 °C and 9.49 °C, respectively. The second model scenario involved filling in beta diversity values for the missing North Atlantic Ocean with values from the Arctic Ocean. This resulted in breakpoints for the 18S and 16S rDNA of 14.4 °C and 12.07 °C, respectively. The third model scenario involved filling in beta diversity values for the missing North

Atlantic Ocean with values from the South Atlantic Ocean. This resulted in breakpoints for the 18S and 16S rDNA of 9.49 °C and 12.22 °C, respectively. The fourth model scenario involved filling in beta diversity values for the missing North Atlantic Ocean with values from both the Arctic Ocean and the South Atlantic Ocean. This resulted in breakpoints for the 18S and 16S rDNA of 14.4 °C and 12.22 °C, respectively.

A break-point analysis was performed for the Pfam protein families beta diversity against temperate with the North Atlantic Ocean samples (Stratiphyt-II) removed to test whether key results remain unchanged (Supplementary Fig. 10e). A breakpoint of 18.2 °C was determined with a p-value of 1.65e−11. Hence, the main result (Fig. 5A) remains unchanged.

**IPCC-based modelling of geographical shifts in beta-diversity breakpoints across the North Atlantic**. To assess where these boundaries are, we began with the HadISST dataset[93], taking the 1961–1990 climatology (Fig. 6). For estimates of changes over the 21st century, we used the RCP 8.5 HadGEM2-ES CMIP5 experiment[37]. A historical HadGEM2-ES experiment was also run for CMIP5, which we used to bias-correct the projected temperatures. This was achieved by determining the differences between the 1961–1990 HadISST and HadGEM2-ES temperatures for each grid box and adding them to the projections. Grid boxes that contain sea ice in the climatology are ignored from this analysis.

**Reporting summary**. Further information on research design is available in the Nature Research Reporting Summary linked to this article.

## Data availability

iTAG rDNA Data: https://opendata.earlham.ac. Eukaryotic metatranscriptome data: https://genome.jgi.doe.gov/. (https://doi.org/10.25585/1488054).

Published online: xx xxx 2021

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

## Acknowledgements

We would like to thank Captains Schwarze and Wunderlich and the RV 'Polarstern' crews of the ARK27-1, ANT29-1 and PS103 expeditions for their vital help during sampling. The Southern Ocean sampling was performed as part of project AWI_PS103_04. The work conducted by the U.S. Department of Energy Joint Genome Institute, a DOE Office of Science User Facility, is supported by the Office of Science of the U.S. Department of Energy under contract no. DE-AC02-05CH11231. R.M.L. acknowledges funding from BBSRC Core Strategic Programme Grant BB/CSP1720/1. T.M. acknowledges funding from the U.S. Department of Energy, Joint Genome Institute (Grant 532, Community Science Program) and the Natural Environment Research Council (NERC) (Grants NE/K004530/1; NE/R000883/1). The PhD studentship of K.M. was funded by the University of East Anglia (UEA) and the Earlham Institute. The PhD studentship of K.S. was funded by the School of Environmental Sciences at UEA.

## Author contributions

T.M. conceived and coordinated the project and helped with sample preparation and data analysis. The data analysis was directed by T.M. and V.M. in cooperation with R.M.L. T.M. wrote the manuscript with contributions from K.M., K.S. V.M., R.M.L, and A.T. K.S. collected samples in the Arctic and South Atlantic with help of M.G. and M.R.R., performed nucleic acid extractions. K.M. performed the analysis of most sequence data generated by JGI. A.T. contributed to sequence analysis performed by K.M. C.P.D.B., K.T. and WvdP provided samples from the Stratiphyt Cruise (22 stations between Canary Islands and Iceland). A.F., K.U.V. and B.B. provided samples from the Southern Ocean and performed sample preparation for sequencing. C.A.B. performed the IPCC-based modelling under the guidance of T.M.L. E.L., K.B., A.C., C.G.D., Bri.F., Bry.F. M.H., K.P., T.B.K.R., N.N.I., N.C.K., S.M. and N.V. performed library preparation, sequencing and IMG-based analyses at JGI. I.V.G., S.R., S.G.T. and E.E.-F. coordinated sequencing work at JGI.

## Competing interests

The authors declare no competing interests.
