## [Peer Review File · Nature Communications]

Peer Review File

Reviewer comments, first round -

Reviewer #3 (Remarks to the Author):

The work covers a large geographic area and generates a huge amount of data. I also appreciate the novel insights into the breakpoint.

As minor comments, I would like to see 1) a brief but clear description about how genome sizes of eukaryotic microbes were estimated, and 2) original data used in the 18S copy number vs genome made available in a database because this is going to be useful for future research. In light of one of the other reviewers' comments about what is new compared to recent studies using similar methods (but different sampling scheme), I suggest that authors provide a concluding paragraph to clearly and concisely tell readers what findings are confirmatory and what are novel.

Reviewer #4 (Remarks to the Author):

General comments:

The paper could benefit from some clearer analysis and figure-making. At the moment it is hard to read the results as they are presented and there are some crucial details missing. The laboratory methods are very dissimilar between the samples, which is something that has to be addressed in the analysis prior to any other analyses.

The network analysis is not necessarily meaningful in the way it is presented. The WGCNA is a great tool that allows you to identify strong relationships between taxa or genes, however, the whole networks are not presented in this manuscript. It would be nice to see all taxa/genes visualized in a single network to see the warm/cold clustering.

There is some clean-up that needs to be done in the figures. They are a little messy and seem quite bunched up, especially figure 3 and some of the extended data figures (less important). The colors on the warm and cold networks should be red and blue throughout the manuscript. i.e. there should be a red box around the warm network in figure 2, and so on...

It would be nice if the whole paper is streamlined a little because of the three different datasets analyzed, it gets very hard to know exactly what is being shown when (and it doesn't help to not have the figure legends on the same pages as the figures).

I would suggest the authors read Djurhuus et al. 2020 in Nature communications:

<https://www.nature.com/articles/s41467-019-14105-1#Sec18>

This paper has quite a few similarities to the presented paper.

Introduction:

Generally well written.

Methods:

The sampling seems a bit strange to me. Samples are not all pre-filtered and there are two different pore sizes that the samples are filtered onto. This can be a huge source of variation. As far as I can tell, samples were not collected in the same fashion from any of the oceans. I would like to see a multivariate analysis of these samples to double-check that the samples from different cruises, but in the same-ish area, are even similar in community structure. (After reading further I found the CCA, which shows what I wanted to see. However, I would still like to see these results for 16S and 18S).

Again for the RNA and DNA extraction, there are different methods utilized and again here it would be nice to see if the samples that theoretically should be similar really are.

L 840: I do not see these two networks anywhere?

L 846: And this network?

Results:

I am not sure if I understand the first header on line 140.

L 146: Were there only 82 samples collected? There were no triplicates for 16S or 18S sequencing or analysis? Any negatives? Positives?

L 208: I don't understand how the temperature was shown to be the primary driver? The other environmental variables also have significant correlations to these networks.

Figure 1:

The Phosphate and silicate should probably be plotted onto a different scale to actually see the variation.

Figure 2:

This figure is not very interesting per se, where did the cold and warm networks come from? It would be more meaningful to make a network of all the data to see the actual split (clustering) between the warm and cold environments. After that, it would be nice to see the different networks separately. Are all the Pfams plotted in this network significantly correlated? I cannot find an edge weight threshold? Maybe it could be an idea to only show the most important connections?

Figure 4:

I am not sure it makes sense to correlate the networks with lat and long. How can it be correlated to both of these? Maybe latitude makes sense, but not necessarily longitude.

I am also not convinced depth makes sense. This would only really be correlated as a function of the chlorophyll a max being deeper because of location.

Point-by-point response (**blue**) to reviewer comments (**black**) for “*The biogeographic differentiation of algal microbiomes in the surface ocean from pole to pole.*” by Martin et al.

Text copied from revised manuscript in *italics*.

REVIEWER COMMENTS

Reviewer #3 (Remarks to the Author):

The work covers a large geographic area and generates a huge amount of data. I also appreciate the novel insights into the breakpoint.

As minor comments, I would like to see 1) a brief but clear description about how genome sizes of eukaryotic microbes were estimated,

A more detailed and clear description of how the genome size of eukaryotic microbes were established has been added to the methods part of our paper (*italics* for new text):

Normalisation of 18S rDNA gene-copy number

18S rDNA gene copy number varies widely among eukaryotes. In order to create an estimate of abundances of the species in the samples the data had to be normalised. Previous work has explored the link between copy number and genome size (Prokopowich *et al.*, 2003). However, there is not a single database of 18S rDNA gene copy numbers for eukaryote species. In order to address this, gene copy number and related genome sizes of 185 species across the eukaryote tree was investigated and plotted (Extended data figure 2) (Godhe *et al.*, 2008), (Carlton *et al.*, 2013), (Torres-Machorro *et al.*, 2010), (Oliver *et al.*, 2007), (Moreau *et al.*, 2012), (Boucher *et al.*, 1991), (Hauser *et al.*, 2010), (Prokopowich *et al.*, 2003), (Rödström, 2017), (NCBI Resource Coordinators, 2016) and (Nordberg *et al.*, 2014). Based on the log transformed data, a significant correlation with a R² of 0.55 with a p-value < 2.2e-16 between genome size and 18S copy number was observed. A regression equation was determined ($f(x)=0.66X+0.75$) as shown in Extended data Figure 2.

To derive this equation, the genome sizes for the species in the reference datasets were retrieved from the NCBI genome database. Since some of the genome sizes were unavailable, for species with missing genome sizes, an average of available genome sizes in closely related species was taken instead. More specifically, first a taxonomic lineage of the relevant subset of the NCBI database was obtained by submitting the taxa IDs using the NCBI taxtastic tool (NCBI Resource Coordinators, 2016). Average genome sizes were then calculated by utilizing the parent ID and taxa ID columns and the known genome sizes of the lowest common ancestor. The 18S datasets were normalised by assigning their genome sizes using the regression equation. The files were further normalised by applying the hits per million reads method.

and 2) original data used in the 18S copy number vs genome made available in a database because this is going to be useful for future research.

The data used to generate the equation of the line has been made available in Supplementary_data_4. We are glad to hear that this is considered to be useful information.

In light of one of the other reviewers' comments about what is new compared to recent studies using similar methods (but different sampling scheme), I suggest that authors provide a concluding paragraph to clearly and concisely tell readers what findings are confirmatory and what are novel.

Thanks for this suggestion. We have added the following paragraph to inform the reader which findings are confirmatory and which are novel.

Taken together, our study confirms the latitudinal distribution pattern in local (alpha) diversity of complex marine microbial communities with a significant decrease from the equator towards polar ecosystems (Extended data figure 7) (e.g., Ibarbalz et al. 2019). However, pole-to-pole datasets offer the opportunity to identify what drives differences in the variation of alpha diversity across larger biogeographic regions (beta diversity). The latter, to the best of our knowledge, has never been estimated before for oceanic microbes although this knowledge is instrumental for spatial scaling of changes in diversity, i.e., loss and gain (e.g., Mori et al. 2018). The application of beta diversity to pole-to-pole algal microbiomes revealed for the first time that physico-chemical differences between polar and non-polar surface oceans have the strongest influence not only on changes in their diversity but also the gene expression activity of their primary producers. Consequently, there appears to be an ecological boundary in sub-polar oceans, which alters the spatial scaling of algal microbiomes (break points in beta diversity) and therefore acts as a gate, shifting poleward due to global warming.

Reviewer #4 (Remarks to the Author):

General comments:

The paper could benefit from some clearer analysis and figure-making. At the moment it is hard to read the results as they are presented and there are some crucial details missing. The laboratory methods are very dissimilar between the samples, which is something that has to be addressed in the analysis prior to any other analyses.

The network analysis is not necessarily meaningful in the way it is presented. The WGCNA is a great tool that allows you to identify strong relationships between taxa or genes, however, the whole networks are not presented in this manuscript. It would be nice to see all taxa/genes visualized in a single network to see the warm/cold clustering.

Thanks for these suggestions. We agree and have updated the Pfam and 18S/16S networks to show the warm/cold clustering. Please see below and in the manuscript for figures 2a and 4a. Furthermore, Supplementary_data_2 and _5 contain information about the number of connections for 18S/16S (all nodes) and Pfams (top 100 nodes), respectively.

Figure 2a. Pfam WGCNA subnetworks. Please see manuscript for details.

Figure 4a. 18S/16S WGCNA subnetworks. Please see manuscript for details.

There is some clean-up that needs to be done in the figures. They are a little messy and seem quite bunched up, especially figure 3 and some of the extended data figures (less important). The colors on the warm and cold networks should be red and blue throughout the manuscript. i.e. there should be a red box around the warm network in figure 2, and so on...

We have done our best to clean-up the figures. However, there are some constraints especially by WGCNA as this reviewer might be aware of. They cannot be overcome unless we change the code of the function, which we do not intend to do for obvious reasons. We have to assume that this reviewer referred to figure 4 because figure 3 is just composed of 2 panels. Our redesign of figures 2 and 4 was done with this suggestion in mind. For instance, we removed some panels (e.g. hair balls) to declutter these figures. Furthermore, identical colours and colour scales have been used throughout to minimise misunderstandings and to improve clarity within the constraints given by WGCNA. Considering the size and complexity of our dataset, we think our new figures 2 and 4 have much improved in terms of clarity.

It would be nice if the whole paper is streamlined a little because of the three different datasets analyzed, it gets very hard to know exactly what is being shown when (and it doesn't help to not have the figure legends on the same pages as the figures).

I would suggest the authors read Djurhuus et al. 2020 in Nature communications: <https://www.nature.com/articles/s41467-019-14105-1#Sec18>
This paper has quite a few similarities to the presented paper.

Many thanks for the link to this paper. It has helped us to streamline our manuscript. We hope that our revised figures 2 and 4 have addressed the reviewers concerns. Please consider the size of our dataset and the complexity of the analyses when judging on the presentation of our data.

Introduction:
Generally well written.

Methods:
The sampling seems a bit strange to me. Samples are not all pre-filtered and there are two different pore sizes that the samples are filtered onto. This can be a huge source of variation. As far as I can tell, samples were not collected in the same fashion from any of the oceans. I would like to see a multivariate analysis of these samples to double-check that the samples from different cruises, but in the same-ish area, are even similar in community structure. (After reading further I found the CCA, which shows what I wanted to see. However, I would still like to see these results for 16S and 18S).

We agree. To address this concern, we have added a PCoA for the 16S and 18S datasets. See below and **Extended_Data_figure_6c, d**. Just to clarify, there was only one expedition (North Atlantic Stratiphyt-II, see M+M) for which a slightly different sampling protocol was used. However, the CCA together with our new 16 and 18S PCoA analyses provide evidence that the slightly different sampling protocols have not introduced variation that seem to have impacted the overall results of our study.

B

	Temperature	Salinity	Nitrate/Nitrite	Phosphate	Silicate
Metatranscriptome Pfam dataset (n=79):	0.08025	0.04722	0.07495	0.07785	0.07503
16S rDNA dataset (n=57):	0.142	0.08499	0.03084	0.0663	0.05178
18S rDNA dataset (n=54):	0.1084	0.0827	NA	0.0544	0.04135

Extended Data figure 6. A PCoA analysis for 16 and 18S have been added to this figure (6c, d). For details, please see manuscript.

L 840: I do not see these two networks anywhere?

L 846: And this network?

Sorry about that. We have updated the figure numbers in the paper in the methods section, under the heading “Network analysis”, on L880 and L886.

Results:

I am not sure if I understand the first header on line 140. (amend to make it clearer)

To be honest, we do not know how to make it clearer: "***A new meta-omics resource for algal microbiomes in the surface ocean from pole to pole.***" This header is for introducing our new data set as a new resource to be exploited by colleagues and their research questions. It makes sure that readers understand that our work is not based on any existing meta-omics dataset, i.e., Tara Oceans. We think this is important because Tara Oceans data are reused and published even in high-impact factor journals.

L 146: Were there only 82 samples collected? There were no triplicates for 16S or 18S sequencing or analysis? Any negatives? Positives? Pros and cons (discuss)???

This is the first pole-to-pole dataset for marine phytoplankton metatranscriptomes. Furthermore, we believe that more is not always better. The application of beta diversity to pole-to-pole algal microbiomes revealed for the first time that physico-chemical differences between polar and non-polar surface oceans have the strongest influence not only on changes in their diversity but also the gene expression activity of their primary producers. This overall novel result would not change even if we would sequence 10x as many samples from pole-to-pole. Yet, it would lead to refinements of the current results. However, the vast majority of scientific studies are constrained by the funds that are available. Nevertheless, triplicate sequencing would have provided insights into reproducibility, but usually reproducibility of sequence data obtained from the same community is high (e.g. Tsementzi et al., 2014, *Env. Micro. Rep.* 6:640). Furthermore, as our sequence data were analysed using validated and standardized tools (e.g. Chen et al., 2020, *Nuc. Acids Res.* 49:D751), we believe that our results are highly reproducible, too.

L 208: I don't understand how the temperature was shown to be the primary driver? The other environmental variables also have significant correlations to these networks.

To address this question, we reiterate what we have said in our manuscript and which refers directly to the output of WGCNA and therefore our network analysis: "*A correlation statistical analysis which is part of the WGCNA package was conducted. This involved taking each network's 'eigengene', a term used by WGCNA, which is **the first principal component of a network, to be representative of that network** in order to conduct a correlation analysis of networks to the environmental variables as shown in Figure 2c. Based on this work, **temperature was identified as the primary driver for both networks**,...*". Thus, temperature was identified by the correlation analysis as the primary driver. Other variables are also significantly correlated as this reviewer rightly pointed out but according to the analysis, they cannot be considered the primary driver. In an ecological context, this makes a lot of sense considering the influence of temperature on water column stratification and mixing and therefore differences in nutrient concentrations, for instance. Please see our discussion in the manuscript for details.

Figure 1:

The Phosphate and silicate should probably be plotted onto a different scale to actually see the variation.

Thanks for this suggestion. We have revised figure 1 accordingly. See below.

Figure 1. D = silicate; E = phosphate. See manuscript for more details.

Figure 2:

This figure is not very interesting per se, where did the cold and warm networks come from? It would be more meaningful to make a network of all the data to see the actual split (clustering) between the warm and cold environments.

See above for new figures to address this suggestion.

After that, it would be nice to see the different networks separately. Are all the Pfams plotted in this network significantly correlated? I cannot find an edge weight threshold?

Thanks for this suggestion. It has been addressed earlier. See above. Just to reiterate, we have removed the hair balls and instead show the actual clusters, which are more meaningful. Furthermore, **Supplementary_data_2 and _5** contain information about the number of connections for 18S/16S (all nodes) and Pfams (top 100 nodes), respectively.

Figure 4:

I am not sure it makes sense to correlate the networks with lat and long. How can it be correlated to both of these? Maybe latitude makes sense, but not necessarily longitude. I am also not convinced depth makes sense. This would only really be correlated as a function of the chlorophyll a max being deeper because of location.

We appreciate these suggestions. However, a previous reviewer (1#) at "*Nature Microbiology*" suggested more emphasis on stratification (Previous reviewer 1#: "*I also think they should add some analyses of stratification across their sites.*"). Hence, latitude and depth were important to this reviewer and we therefore had to consider those parameters. This suggestion therefore has shaped the current manuscript. Thus, it seems that there are different opinions in terms of which environmental variables should be included. This might be driven by different interests and different scientific expertise of readers. With that in mind, we think that showing all of these environmental variables might be the best solution because a) it reflects a transparent approach of data sharing and b) it might address different questions and interests. The latter might therefore help subsequent studies that wish to build on our manuscript.

Reviewer comments, second round –

Reviewer #5 (Remarks to the Author):

This study analyzes an extraordinary set of amplicons and metatranscriptomic data from samples collected at chlorophyll maxima and covering the Atlantic Ocean from pole to pole. The authors carried out a series of interesting statistical analyzes to discern the main factors involved in shaping the basin-scale biogeography of eukaryotic phytoplankton. The analyzes made it possible to establish (geographic/environmental) break-points that differentiate the structure and gene expression patterns of the predominant algal microbiomes. The dataset is excellent and, in fact, this is my main concern: it would deserve more thorough analysis and interpretation. The analyzes presented indicate that temperature is the most important factor controlling phytoplankton biogeography. However, even though seawater temperature is strongly correlated with nutrient availability – the authors actually cite this negative relationship in the manuscript, they do not provide evidence to support one factor over the other, even though temperature and nutrients control the ecophysiology of phytoplankton through quite different mechanisms (i.e., different temperature optima and different nutrient uptake kinetics, involving different core sets of genes and proteins). How could we say that temperature is the main factor without separating the effect from other co-operating factors? Statistical analyzes hardly separate the effects of temperature and nutrients. In fact, although temperature and nutrients are typically anticorrelated, their variability is actually a bit more complex. Note that high nutrient concentrations may simply reflect the early stages of a nutrient pulse and vice versa, the lack of nutrients in the medium could simply reflect rapid biological uptake, that is, nutrient supply and biological consumption are dynamic processes. Although organisms have their own temperature optimal and tolerance ranges, they also compete for limiting nutrients, and the outcome of this competitive process is highly dependent on the number and turnover rate of membrane transporters. Since this manuscript presents temperature (the latitudinal temperature gradient) as the most relevant factor in establishing the biogeography and breakpoints of beta diversity of algal microbiomes, would it be possible to explore the relative influences of temperature and nutrients using metatranscriptomic data? Otherwise, the study's conclusions that temperature per se is the main factor would be quite speculative. Alternatively, both factors may be relevant and the manuscript approach should be reconsidered.

The authors state throughout the article that their analyzes examine the algae microbiomes of the ocean surface, but this would not be the case considering that chlorophyll maximum layers are often located at the bottom of the euphotic zone.

I add here my review of the authors comments to Reviewer 4.

[REVIEWER 4] The laboratory methods are very dissimilar between the samples, which is something that has to be addressed in the analysis prior to any other analyses.

This is not commented on in the response letter or addressed in the manuscript and is relevant.

[REVIEWER 4] L 146: Were there only 82 samples collected? There were no triplicates for 16S or 18S sequencing or analysis? Any negatives? Positives? Pros and cons (discuss)???

This question about the reproducibility of the 18s 16s data has been adequately justified by the authors.

[REVIEWER 4] In line with my concern about the relative effects of temperature and nutrient supply – the reviewer also raised this concern arguing that variables other than temperature also exhibited strong correlations. The authors replied to this question: "Other variables are also significantly correlated as this reviewer rightly pointed out but according to the analysis, they cannot be considered the primary driver."

The authors should explain clearly why.

[REVIEWER 4] The authors also say: "In an ecological context, this makes a lot of sense considering the influence of temperature on water column stratification and mixing and therefore differences in nutrient concentrations, for instance."

But in fact, this is the question, temperature is (anti)correlated with other factors, the most critical being nutrient availability and therefore why is assumed to be temperature and not nutrients the main driver?

[REVIEWER 4] Figure 1: The Phosphate and silicate should probably be plotted onto a different scale to actually see the variation.

I assume that temperature and nutrient data are referred to as surface waters. I would ask the authors to provide a figure showing the hydrographic structure of the water column along the latitudinal gradient – for example, the upper mixed layer depth or the nutricline depth, otherwise the surface concentrations of nutrients say little about the nutrient availability for the communities inhabiting the chlorophyll a maximum layers of the low latitudes. These communities are often strongly dependent on the depth of the nutricline and the magnitude of the nutrient gradient across the nutricline. Taking into account that DCM are widespread in low latitude oceans, it seems rather strange to do a study on the biogeography of the phytoplankton communities inhabiting chlorophyll a maximum layers showing data on the hydrographic variability of surface waters,

[REVIEWER 4] Figure 2: This figure is not very interesting per se, where did the cold and warm networks come from? It would be more meaningful to make a network of all the data to see the actual split (clustering) between the warm and cold environments.

This question has been adequately addressed by the authors in the new version of the ms.

Point-by-point response to comments from reviewer 5 on NCOMMS-2105933A
(Black- reviewers comment; blue – our response including new data)

REVIEWER COMMENTS

Reviewer #5 (Remarks to the Author):

Comment: The dataset is excellent and, in fact, this is my main concern: it would deserve more thorough analysis and interpretation. The analyzes presented indicate that temperature is the most important factor controlling phytoplankton biogeography.

Response: We are glad this reviewer is excited about our dataset. It is the outcome of many years (>8 years) of collaborative work.

Comment: However, even though seawater temperature is strongly correlated with nutrient availability – the authors actually cite this negative relationship in the manuscript, they do not provide evidence **to support one factor over the other**, even though temperature and nutrients control the ecophysiology of phytoplankton through quite different mechanisms (i.e., different temperature optima and different nutrient uptake kinetics, involving different core sets of genes and proteins). How could we say that temperature is the main factor without separating the effect from other co-operating factors? Statistical analyzes hardly separate the effects of temperature and nutrients. In fact, although temperature and nutrients are typically anticorrelated, their variability is actually a bit more complex. Note that high nutrient concentrations may simply reflect the early stages of a nutrient pulse and vice versa, the lack of nutrients in the medium could simply reflect rapid biological uptake, that is, nutrient supply and biological consumption are dynamic processes. Although organisms have their own temperature optimal and tolerance ranges, they also compete for limiting nutrients, and the outcome of this competitive process is highly dependent on the number and turnover rate of membrane transporters. Since this manuscript presents temperature (the latitudinal temperature gradient) as the most relevant factor in establishing the biogeography and breakpoints of beta diversity of algal microbiomes,

Response: We would like to clarify this with our response because our manuscript provides evidence “to support one factor over the other”. However, we agree that it is indeed difficult to disentangle the effects of temperature and nutrients on the ecophysiology and diversity of phytoplankton as both variables can influence phytoplankton on different scales. On a cellular level, temperature impacts enzyme activity and therefore fundamental traits such as cell division whereas on the ecosystems level, temperature impacts the stability of the water column, which has consequences for the availability of nutrients. Furthermore, the size of individual phytoplankton cells has an impact on the demand of nutrients to support cell division and therefore growth. To address all these intertwined processes and dependencies with meta-omics largely is impossible as the sequence space is highly dimensional (e.g., many different genes in a complex cellular and environmental context). Thus, most meta-omics studies reduce the number of dimensions by applying correlative approaches such as PCA, MDS, and CCA, which help to ordinate complex data sets and therefore enable to compare variables with one another such as temperature and diversity. Statistics will provide additional information about how robustly the variability of a data set (e.g., diversity) can be explained by certain environmental variables such as temperature or specific nutrients. Thus, our analyses neither provide causal nor mechanistic insights into how temperature and nutrients influence the observed changes in beta diversity, which agrees with the comment that “*their variability is actually a bit more complex*”. If we look at the different correlations and their

associated significances, though, a larger picture emerges in which temperature appears to play an important role especially for identifying the break points in beta diversity (see below).

Comment: would it be possible to explore the relative influences of temperature and nutrients using metatranscriptomic data?

Response: We would like to reiterate the evidence (Three independent analyses) we presented in the current version of the manuscript for the important role of temperature in the context of “would it be possible to explore the relative influences of temperature and nutrients using metatranscriptomic data?”

1) **First piece of evidence:**

This is text (red) copied from the latest version of our MS. We don't know how to be even clearer about our results: “To identify which environmental variable was most responsible for a possible latitudinal differentiation in gene co-expression networks, we applied a weighted gene co-occurrence network analysis (WGCNA) (Langfelder & Horvath 2008) based on Pfam gene counts. Our WGCNA revealed that there were two gene co-expression networks only based on positive links (Figure 2a, supplementary data 2). A correlation statistical analysis which is part of the WGCNA package was conducted. This involved taking each network's ‘eigengene’, a term used by WGCNA, which is the first principal component

of a network, to be representative of that network in order to conduct a correlation analysis of networks to the environmental variables as shown in Figure 2b. Based on this work, temperature was identified as the primary driver for both networks, which corroborates results from our CCA analysis (See above and extended data figure 6). Whereas salinity was co-correlated with temperature, the major inorganic nutrients such as nitrate, phosphate and silicate were significantly ($p\text{-value} \leq 0.001$) anti-correlated to temperature and salinity. The gene co-expression network designated as blue ($N = 1614$ Pfams) has a strong positive relationship with temperature (correlation coefficient of +0.72; $p\text{-value} = 2e\text{-}12$), hence, this is considered to be the warm network. The network designated as turquoise ($N = 2369$

Fig. 2b. Copied from MS. It shows all env. variables including their correlations with the subnetworks. Thus, we think this addresses the main remark of reviewer 5. Furthermore, these results are corroborated by the CCA analysis (Extended fig. 6) and breakpoints in beta diversity (Extended figs. 9 & 10).

Pfams) has a strong negative relationship with temperature (correlation coefficient of -0.8; p-value = 1e-16), hence, this is considered to be the cold network.”

2) **Second piece of evidence:**

Extended data figure 6: A) CCA ordination plot of metatranscriptome samples (based on their Pfam protein domain content) with fitted environmental variables. B) Table of percentage variation attributed to each environmental variable from CCA analyses for 18s, 16s and metatranscriptome data sets. In each case, temperature accounts for the highest percentage of variation. This corroborates results shown in figure 2b (see above).

B

	Temperature	Salinity	Nitrate/Nitrite	Phosphate	Silicate
Metatranscriptome Pfam dataset (n=79):	0.08025	0.04722	0.07495	0.07785	0.07503
16S rDNA dataset (n=57):	0.142	0.08499	0.03084	0.0663	0.05178
18S rDNA dataset (n=54):	0.1084	0.0827	NA	0.0544	0.04135

Extended data figure 6. A) CCA plot. **B)** Represents a table of CCA on each dataset of 18S, 16S and metatranscript Pfam against the individual environmental variables. The numbers in the table are the percentage of the variation that each variable accounts for in each dataset.

3) **Third piece of evidence:** The

breakpoint analysis of the beta diversity data with respect to all measured environmental variables (Extended figure 10) has revealed that only temperature caused breakpoints. We could not perform breakpoints with the other environmental variables (See extended data figure 10).

Taken together, we have provided detailed statistical analyses for the relative influence of temperature and nutrients by using not only the metatranscriptome data but also the 16 and 18S rDNA data. Thus, we believe this addresses the main remark of reviewer 5 unless this reviewer has a specific method in mind to be applied in addition to our 3 different and independent approaches to discuss the role of all environmental variables with respect to their influence on the identified subnetworks. However, we believe that additional statistical methods will lead to corroborating results. Furthermore, we agree that any analyses which will require the generation of new data are beyond the scope of a revision. They should be subject to subsequent work.

Comment: Otherwise, the study's conclusions that temperature per se is the main factor would be quite speculative. Alternatively, both factors may be relevant and the manuscript approach should be reconsidered.

Response: Despite the independent evidence we have provided for the role of temperature (see above), to acknowledge the correlative nature of our analyses and the physiological complexity underlying the covariance between temperature and nutrients, we have toned down the role of temperature in shaping beta diversity and the co-occurrence networks throughout our manuscript. Please see manuscript version with tracked changes.

Comment: The authors state throughout the article that their analyzes examine the algae microbiomes of the ocean surface, but this would not be the case considering that chlorophyll maximum layers are often located at the bottom of the euphotic zone.

Response: We have replaced “surface ocean” with “upper ocean”. The latter term includes the bottom of the euphotic zone including the chlorophyll maximum layers (e.g. Sprintall & Cronin, 2009, Encyclopedia of Ocean Sciences, 2nd ed., pages 217-224).

Comment: I add here my review of the authors comments to Reviewer 4.

[REVIEWER 4] The laboratory methods are very dissimilar between the samples, which is something that has to be addressed in the analysis prior to any other analyses.

This is not commented on in the response letter or addressed in the manuscript and is relevant.

Response: We have addressed that point both in the response letter and manuscript. See below (blue) copied from our response to the reviewers’ letter. However, see page 6 for a sensitivity

Extended data figure 10. Protein families (Pfam) dataset beta diversity plotted against the environmental variables. The numbers correspond to sample locations as shown in figure 1A. The y-axis represents the beta diversity across the stations. The x-axis in panels a, b, c and d represents salinity, silicate, nitrate/nitrite and phosphate. Nutrient concentrations are given in $\mu\text{mol L}^{-1}$, and the unit for salinity is PSU (Practical Salinity Unit).

analysis, providing new additional evidence that the North Atlantic Stratiphyt-II data did not impact the overall results of our study.

“The sampling seems a bit strange to me. Samples are not all pre-filtered and there are two different pore sizes that the samples are filtered onto. This can be a huge source of variation. As far as I can tell, samples were not collected in the same fashion from any of the oceans. I would like to see a multivariate analysis of these samples to double-check that the samples from different cruises, but in the same-ish area, are even similar in community structure. (After reading further I found the CCA, which shows what I wanted to see. However, I would still like to see these results for 16S and 18S).”

We agree. To address this concern, we have added a PCoA for the 16S and 18S datasets. See below and **Extended_Data_figure_6c, d**. Just to clarify, there was only one expedition (North Atlantic Stratiphyt-II, see M+M) for which a slightly different sampling protocol was used. However, the CCA together with our new 16 and 18S PCoA analyses provide evidence that the slightly different sampling protocols have not introduced variation that seem to have impacted the overall results of our study.

Extended Data figure 6. A PCoA analysis for 16 and 18S have been added to this figure (6c, d). For details, please see manuscript.

To provide additional evidence that the North Atlantic Stratiphyt-II data did not impact the overall results of our study, we conducted **a)** an outlier analysis and **b)** completely removed the North Atlantic data and redid our breakpoint analysis with the metatranscriptome data to see if the main results stay the same.

A) Outlier analysis: We decided to use the PCA data for this analysis as an appropriate approximation for the other ordination methods we applied (e.g., CCA). We calculated PCA with a function called `prcomp` from the R package called `stats` (<https://www.r-project.org>). We examined the PC1 and PC2 for outliers. To calculate the Mahalanobis distance, firstly we calculate a covariance matrix of distribution with a function called `covRob` with the parameter “`estim`” set to `pairwiseGK` from the R package called `robust` (<https://cran.r-project.org/web/packages/robust/index.html>).

Using these distances, we applied the function `pchisq` from the R package called `stats` to generate a chi-squared distribution and corresponding p-values. Any outlier samples were identified with a p-values < 0.05.

18S:

PC1 % variance -> 29.9%

PC2 % variance -> 18.67%

Outliers -> **29, 34, 37, 52,**

All from North Atlantic Stratiphyt-II.

16S:

PC1 % variance -> 30.95%

PC2 % variance -> 10.79%

Outliers -> **None found**

MetaT:

PC1 % variance -> 22.4%

PC2 % variance -> 6.645%

Outliers -> **None found**

Thus, there were only 4 outliers identified for the 18S data, which corresponds to 7.4% of all 18S data in our manuscript. We have not removed them because we wanted to be as inclusive as possible with our data and to comply with general data policy and transparency regulations. For the other two data sets, there were no outliers identified. Thus, it is very likely that the North Atlantic Stratiphyt-II data and therefore the different laboratory methods did not significantly influence the main outcome of our study.

B) Removal of North Atlantic Stratiphyt-II data to test whether key results remain unchanged:

We removed all metatranscriptome data from the North Atlantic Stratiphyt-II cruise and redid the beta diversity breakpoint analysis exactly as down previously. In spite of removing a significant amount of data, a breakpoint of **18.20°C** was determined with a p-value of 1.651e-11 (See figure down below on page 7). The temperature for the breakpoint which included the North Atlantic Stratiphyt-II data was estimated to be at **18.06°C**. Hence, the main result remains unchanged.

Comment: [REVIEWER 4] In line with my concern about the relative effects of temperature and nutrient supply – the reviewer also raised this concern arguing that variables other than temperature also exhibited strong correlations. The authors replied to this question: “Other variables are also significantly correlated as this reviewer rightly pointed out but according to the analysis, they cannot be considered the primary driver. “

The authors should explain clearly why.

Response: See above please and toned-down version of our manuscript with tracked changes.

[REVIEWER 4] The authors also say: “In an ecological context, this makes a lot of sense considering the influence of temperature on water column stratification and mixing and therefore differences in nutrient concentrations, for instance.”

But in fact, this is the question, temperature is (anti)correlated with other factors, the most critical being nutrient availability and therefore why is assumed to be temperature and not nutrients the main driver?

Response: See above please and toned-down version of our manuscript with tracked changes.

[REVIEWER 4] Figure 1: The Phosphate and silicate should probably be plotted onto a different scale to actually see the variation.

I assume that temperature and nutrient data are referred to as surface waters. I would ask the authors to provide a figure showing the hydrographic structure of the water column along the latitudinal gradient – for example, the upper mixed layer depth or the nutricline depth, otherwise the surface concentrations of nutrients say little about the nutrient availability for the communities inhabiting the chlorophyll a maximum layers of the low latitudes. These communities are often strongly dependent on the depth of the nutricline and the magnitude of the nutrient gradient across the nutricline. Taking into account that DCM are widespread in low latitude oceans, it seems rather

strange to do a study on the biogeography of the phytoplankton communities inhabiting chlorophyll a maximum layers showing data on the hydrographic variability of surface waters,

Response: We have plotted the nutrients on a different scale as requested by reviewer 4. Hence, we have completely addressed this comment. Reviewer 5 suggests new analyses that are beyond the scope of this revision.